# Interventional Oncology for Colorectal Liver Metastases: From Local Cure to Salvage Therapy

**DOI:** 10.3390/biomedicines13092182

**Published:** 2025-09-06

**Authors:** Dimitrios Xenos, Vlasios S. Sotirchos, Platon M. Dimopoulos, Constantinos T. Sofocleous

**Affiliations:** 1Interventional Radiology Service, Memorial Sloan Kettering Cancer Center, New York, NY 10065, USA; xenosd@mskcc.org (D.X.); sotirchv@mskcc.org (V.S.S.); 2Department of Interventional Radiology, University Hospital of Patras, 26504 Patras, Greece; dimopoylos.platonas@gmail.com

**Keywords:** colorectal cancer, interventional oncology, liver metastases, locoregional therapy, margin, thermal ablation, Yttrium-90 radioembolization, chemoembolization

## Abstract

Cancer is a leading cause of cancer-related death. Liver metastases develop in over one-third of patients and are associated with worse prognosis. The evolution in the field of interventional oncology/radiology over the past two decades has expanded image-guided locoregional therapies for colorectal liver metastases (CLM). Historically, hepatic resection was considered the only possible cure for selected patients with CLM. Current evidence supports thermal ablation (TA) as another locally curative treatment modality for small CLM that can be ablated with adequate margins. Other non-thermal ablative treatment options include Yttrium-90 (^90^Y) radiation segmentectomy (RS), irreversible electroporation (IRE), and histotripsy, with an evolving role in the treatment of CLM. More extensive disease that is not amenable to resection or ablation can be treated with intra-arterial therapies (^90^Y trans-arterial radioembolization (TARE) and trans-arterial chemoembolization (TACE)). This comprehensive review describes the evolution of interventional oncology treatments for CLM and examines the appropriate indications for each treatment modality.

## 1. Introduction

Colorectal cancer (CRC) ranks as the fourth most frequently diagnosed malignancy worldwide and stands as a leading cause of cancer-related deaths in the United States, especially among younger populations under the age of 50. Among men in this age group, CRC has become the foremost cause of cancer mortality, while it is the second leading cause for women [1]. It is estimated that in 2025, over 150,000 people will be diagnosed with CRC and over 50,000 deaths will be as a result of CRC. Almost half of patients with CRC will present with or develop hepatic metastases that are associated with worse prognosis [2,3,4]. Hepatic resection was historically considered the treatment of choice for colorectal liver metastases (CLM), but approximately 80% of patients are not surgical candidates due to disease extent and/or presence of comorbidities [5].

Technical advances and equipment evolution in the field of interventional radiology over the past two decades have expanded the treatment options for CLM. These image-guided treatments cover a wide range of indications and can be used with local curative intent or offered to manage disease in combination with chemotherapy, especially in the salvage, chemorefractory setting [6,7,8,9,10,11,12,13,14,15,16,17,18,19,20]. Minimally invasive interventional radiology treatments for CLM include percutaneous thermal and non-thermal ablation, trans-arterial radioembolization (TARE) and trans-arterial chemoembolization (TACE) [13,16,21,22,23,24,25,26,27]. Recently a new non-ionizing non-thermal technique using acoustic energy (histotripsy) has demonstrated preliminary promising results [28]. In general, these locoregional treatments provide liver disease control, prevention of liver failure, de-escalation or, whenever feasible, elimination of toxic systemic chemotherapy while prolonging disease-free and overall patient survival, offering a favorable safety profile with low morbidity and preservation of liver parenchyma [29].

The main goal of this comprehensive narrative review is to encapsulate the current clinical evidence regarding the role of interventional oncology/radiology in the management of CLM, as well as to summarize the challenges associated with each type of minimally invasive, image-guided therapeutic approach.

## 2. Ablation Modalities

### 2.1. Thermal Ablation

Radiofrequency ablation (RFA) is the most extensively studied ablative modality for CLM. Either monopolar or bipolar RFA systems generate high-frequency alternating currents of 365 to 500 kHZ, which induce frictional heating with sequential irreversible protein coagulation at 60 to 100 °C and ultimately coagulation necrosis [30]. The main limitation of RFA is the creation of relatively small ablation zones (AZ) and the vulnerability to the “heat-sink” effect [31]. Microwave ablation (MWA) generates electromagnetic waves of frequency between 900 MHz and 2.5 GHz that cause rotation and re-alignment of polar molecules. The concomitant increase in temperature results in protein denaturation and coagulative necrosis. The current evidence favors the latter over RFA, due to its higher efficacy in heat generation and shorter times to reach lethal temperatures, attributed mainly to decreased vulnerability to tissue impedance and to “heat-sink” effects [32]. MWA achieves faster, larger, and more homogenous AZs around the target tumor [33,34].

Cryoablation is a thermal ablation (TA) method that utilizes lethal cold temperature at around −40 °C. The modality causes intracellular ice crystal formation damaging cellular membrane and organelles, inducing cell death by alternate freezing and thawing cycles [35]. An advantage of this method is the direct visualization of the “ice-ball” intra-procedurally with either US, CT, or MRI. Although cryoablation was one of the earliest reported TA modalities, its application for liver tumors has been limited. Early evidence indicated high rates of LTP and severe procedure-related morbidity mediated by the excess release of liver shock cytokines [36,37,38]. The resulting cryo-shock syndrome is characterized by renal failure, disseminated intra-vascular coagulation (DIC), and adult respiratory distress syndrome (ARDS) [36,37,38]. The modality may be useful on tumors located in close proximity to central bile ducts, vessels, or other structures like GI tract [39]. A systematic review and meta-analysis comparing the different local ablation techniques for CLM resulted in a comparable safety profile between the three aforementioned thermal ablation modalities, with MWA associated with statistically significant lower LTP, disease-free survival and overall survival rates in comparison to RFA and cryo-ablation [38]. It is important to note that in half of the included studies the investigators did not assess intra-operatively the minimal ablation margins (MM) of the AZ; most of the studies that included MWA and cryoablation assessed the AZ. RFA constituted the modality that was most commonly utilized without assessing technical completeness.

Percutaneous TA can also be used in the management of CLM through the “test-of-time” approach [40]. The “test-of-time” approach supports that ablation can decrease the number of resections required by achieving complete tumor necrosis in some patients and offering enough time interval to test the tumor biology and detect those that develop new intrahepatic and/or extrahepatic metastases. Therefore, resection can be avoided in patients who are unlikely to gain any survival benefit from the higher risk procedure [40]. Biomarkers for CLM have been proven to be essential in accessing prognosis and achieving better individualized treatment planning [41].

### 2.2. Non-Thermal Ablation

#### 2.2.1. Irreversible Electroporation

Irreversible electroporation (IRE) requires at least two electrodes placed parallel to bracket the target tumor and generate high electric field pulses that lead to cellular membrane disruption with pore formation, resulting in permanent membrane permeation and cell death. The small amount of heat produced during the IRE reaches only sub-lethal doses, leaving the surrounding connective tissue intact. This allows safe ablation of tumors located in close proximity to the portal vein and the bile duct confluence at the liver hilum or other locations deemed at risk for TA [42]. General anesthesia and complete muscle relaxation are obligatory during utilization of IRE to prevent generalized muscle contraction [43]. IRE is efficacious in the management of CLM; however, historically, procedural results have been inferior to TA with regard to long-term local tumor control [23,44,45,46].

#### 2.2.2. Histotripsy

Histotripsy is an image-guided, non-thermal ablation modality. The mechanism of action for acoustic cavitation is the production of focus microsecond ultrasound pulses that sequentially generate gas microbubbles [47]. The pressure threshold is dependent on the intrinsic tension surface of the bubbles and varies between different types of target tissue [48]. The fact that collagen-based tissue require higher energy force to achieve acoustic cavitation (bile ducts, vessels, bowel wall) than the non-collagenous liver parenchyma and tumor allows histotripsy application near critical structures with safety and without any impact from heat-sink effects [49,50,51,52].

In 2019, THERESA trial reported 100% tumor coverage by the created AZ; however, there was residual viable tumor at 1-month post histotripsy imaging in 20% of the cases. The authors reported acceptable safety and efficacy profiles and indicated the need for further investigation of the new treatment modality [53]. #HOPE4LIVER, a prospective international single-arm multi-center phase I/II clinical trial for primary and secondary liver tumors ≤3 cm reported technical success and a complication rate of 95% and 7%, respectively [28,54]. Efficacy and safety outcomes of histotripsy for liver tumors are also currently under investigation by an ongoing phase II clinical trial (NCT06579833 for CLMs ≤ 10 cm) and a single-arm, non-randomized, prospective, observational study (BOOMBOX: Master Study: NCT06486454). Finally, Treatment of Cancer with Immune Checkpoint Inhibition Therapy Boosted by High Intensity Focused Ultrasound Histotripsy (iFOCUS: NCT06524570) constitutes a phase I trial that intends to evaluate the combination of histotripsy and ipilimumab plus nivolumab in the salvage setting for unresectable liver metastases. Safety, tolerance, and feasibility of histotripsy along with radiological, immunological, and clinical response are the primary endpoints of the study.

### 2.3. Evolution of Thermal Ablation as an Equivalent Treatment to Surgical Resection for Colorectal Liver Metastases ≤ 3 cm

Historically, liver resection was the preferred treatment for CLM if all disease was amenable to complete resection. Percutaneous TA was offered to patients rendered non-eligible for surgery for technical reasons, existing comorbidities, or advanced disease with extrahepatic metastases, as well as to treat post-hepatectomy recurrences [55,56,57,58,59,60,61,62,63]. TA with the form of RFA was the first technology widely utilized and investigated in comparison to resection until the development of MWA in 2010. Surgery was favored over TA, as early studies performed in the 2000s demonstrated improved oncologic outcomes with resection [55,64]. Risk factors for local tumor progression (LTP) after resection include suboptimal surgical margins, residual viable tumor, ≥3 target tumors, CLM maximal longitudinal diameter ≥ 5 cm, and carcinoembryonic antigen (CEA) levels > 200 ng/mL [65,66,67,68,69]. The ability to perform pathologic assessment of the surgical margins constituted an objective advantage of surgery over percutaneous ablation. The realization that complete tumor resection with margins impacted oncologic outcomes sparked an interest to assess the results of percutaneous ablation with a similar approach including confirmation of complete ablation with margins [9,70,71,72].

#### 2.3.1. Target Tumor Size

Initially, unresectable CLMs were eligible for ablation without size restriction parameters. The most commonly utilized cut-off value for longitudinal target tumor diameter was 5 cm. Even though some studies were supporting results with survival rates comparable to surgery, ablation efficacy was inferior in terms of tumor control rates [56,57,58,59,60,61,62,63]. An increasing understanding of the limitations of ablation led to the realization of the importance of complete ablation with margins. This meant that the AZ should uniformly cover the target tumor, including an area of at least 5 mm and ideally 10 mm all around the CLM. This is more difficult as the CLM diameter increases. Earlier investigations indicated that technically successful ablation was more common in smaller tumors [73]. Time to local tumor progression (LTP) after percutaneous RFA was inversely associated with CLM size, and survival rates demonstrated significant improvement for tumors with up to 3 cm largest tumor diameter [63,74,75,76]. Based on the increasing evidence showing improved outcomes after ablation of smaller CLM, in 2011 a study that evaluated clinical outcomes for patients undergoing RFA for recurrent CLM in post-hepatectomy settings and an international panel of ablation experts in 2013 recommended decreasing the cutoff for offering percutaneous ablation from 5 cm to 3 cm, ideally to ≤3 tumors, as a locally curative treatment [15,71]. In 2015, a prospective observational RCT MAVERRIC was initiated to prove non-inferiority of MWA in comparison to resection on patients with less than five CLMs with maximal longitudinal diameter of 3 cm. The study was concluded in 2022, resulting in associating ablation with decreased morbidity, shorter time spent in medical facilities, and lower healthcare-related costs within 2 years of the initial treatment, with equal overall survival, highlighting the advantages of ablation for both the patient and the healthcare system [77].

#### 2.3.2. Metastases Location, Number, and Chemotherapy Synergy

Number and location of CLM also constituted a matter of investigation with potential changes that would contribute to the improvement of TA efficacy outcomes. Recommendations were made to modify the initially accepted maximum number from 5 to 3 CLM following evidence of prolonged median survival of the patients with oligometastases [56,71,78]. A smaller number of CLM increases the chance to achieve the desired AZ and preserve a greater percentage of healthy liver parenchyma. Perivascular location of metastases was also associated with higher rates of technically unsuccessful TA. Proximity to a vessel renders all TA modalities prone to heat energy loss (“heat-sink” effect) from the cooling effect of the adjacent blood vessels, especially when the vessel diameter is greater or equal to 3 mm, due to the cooling effect of the flowing blood, thus limiting ablation efficacy and resulting in residual viable tumor that can lead to local progression [31,79]. Older bibliography supported that perivascular CLM location was associated with technical difficulties and worse prognosis in terms of complication and LTP rates [74,75,79]. More recent reports support that ablation is safe and the technical completeness is feasible in perivascular locations especially when using MWA that is less prone to the “heat-sink” effect [33,34,79,80]. More specifically, a retrospective analysis of 110 patients with CLMs treated with percutaneous TA demonstrated that although perivascular tumor location constituted an independent predictor of LTP in the cases treated with RFA, there was no effect for perivascular CLMs when treated with MWA [79].

The addition of percutaneous TA on systemic chemotherapeutic regimens was another effort to improve the clinical outcomes of local curative intent treatments. An RCT supporting the synergy of combining ablation with standard of care chemotherapy resulted in significant improvement in local disease control and survival benefit when compared to chemotherapy alone [81,82]. A 2017 landmark study showed that patients with unresectable CLMs who received RFA (with or without resection) in addition to systemic chemotherapy had improved overall survival compared to those receiving chemotherapy alone (CLOCC trial) [26].

#### 2.3.3. Minimal Ablation Margins Assessment

Another major limitation of TA was the lack of a reliable confirmation method of adequate uniform coverage of the target tumor by the AZ with the desired MM as depicted on cross sectional imaging [70]. In the early 2000s, there was no standardization of the MM evaluation. Ideal timing and assessment methodologies after completion of the ablation procedure were lacking. Initial assessments were based on 2-dimensional (2D) side-by-side comparisons of the pre- and the post-ablation ceCT [9,79,83,84,85,86,87]. A pivotal study evaluating the importance of the MM performed a two-dimensional assessment of the AZ and confirmed the MM constitutes an independent predictive factor of LTP after RFA. The reported reduction in LTP rate risk was 46% for each 5 mm increase in minimal margin size [70]. The authors proposed the immediate assessment of the MM intra-operatively for ablation completeness in addition to the efficacy assessment with a 4 to 8 weeks post-ablation imaging and the standardization of a reliable assessment methodology for margin verification [70]. Following studies demonstrated a statistically significant inverse relationship between LTP and MM ≥ 5 mm and no instances of LTP for tumors treated with MM ≥ 10 mm [79,88]. In *KRAS* mutant CLM, MM ≥ 10 mm are recommended to optimize oncologic outcomes [41,85,86,89,90]. Even for tumors with aggressive biological characteristics, achieving complete tumor ablation, characterized by sufficient margins and verified negative biopsy results from the ablation zone, remains the most significant predictor for effective local control. This comprehensive eradication significantly reduces the risk of local tumor recurrence and is considered the most vital factor in successful treatment control outcomes [91]. The literature supported that intraprocedural MM assessments have greater predictive value on LTP rates in comparison to the historic standard of 4–8 week post-ablation assessements [92,93]. Subsequent investigations indicated that volumetric 3D MM assessments are more reliable and consequently more accurate for predicting risk and location of LTP than the 2D side-by-side comparisons [13]. Specialized software systems allow pre- and post-ablation image registration and 3D evaluation of the AZ and margins (Figure 1) [70,92,93,94,95,96,97,98,99,100]. The use of biomechanical model-based deformable image registration (DIR) software (Morfeus, RayStation, RaySearch Laboratories, Stockholm, Sweden) combined with artificial intelligence autosegmentation has recently been validated as an essential method for confirming ablation completeness in hepatic tumors. A randomized phase II trial demonstrated that this approach accurately quantifies the minimal ablative margin and effectively predicts local tumor control after ablation of primary and secondary liver tumors (COVER-ALL) [101]. A statistically significant superiority in achieving MM ≥ 5 mm when they used this validation method in comparison to visual-only side-by-side AZ assessment was documented [101].

The ongoing international, multicenter, single-arm phase II/III clinical trial ACCLAIM (Ablation with Confirmation of Colorectal Liver Metastasis) is designed to demonstrate that intraprocedural confirmation of a 3D MM of at least 5 mm, or immediate re-ablation when the margin is less than 5 mm, can achieve local tumor control rates exceeding 90%. This trial employs advanced margin confirmation software during MWA of CLM to objectively verify sufficient ablation margins in real time. By mandating a minimal 5 mm margin as the criterion for procedural success and allowing immediate corrective intervention, ACCLAIM aims to establish reproducible technical standards for ablation and improve long-term oncologic outcomes.

#### 2.3.4. Optimization of Ablation Monitoring with Split-Dose PET/CT and CTHA

In addition to the use of CT and ultrasound imaging for target tumor localization, in 2013 the introduction of intraprocedural real-time PET/CT for CLM, employing the split-dose technique, improved tumor detection for targeting. In short, this technique administers the first FDG dose equal to the 1/3 (4 mCi) of the diagnostic dose (4 mCi) within 30–60 min prior to the ablation. This allows tumor localization for targeting, with tumor visibility, typically for 60 to 120 min. Upon completion of ablation, a second administration of FDG amounting to two-thirds of the diagnostic dose (approximately 8 mCi) is administered. This is followed by a repeat PET/CT scan conducted 30 min later. During the interval between the second FDG injection and the image acquisition, a post-ablation ceCT is repeated and employed for AZ and MM assessments. Additionally, confirmatory biopsies of the ablation zone may be performed during this period to enhance validation of ablation completeness [102].

This method offers the significant advantage of allowing continuous visualization of PET-avid tumors through repeated, short breath-hold PET/CT acquisitions in real time, both prior to and during ablation. Real-time PET imaging is capable of detecting most CLMs, including lesions that may not be apparent on non-ceCT scans (Figure 2) [103]. The split-dose PET/CT technique enables precise intraprocedural localization and real-time visualization of the target tumor with electrodes in place, even following hydrodissection or tumor mobilization (Figure 3). This method overcomes the limitations of real-time ultrasound monitoring of the ablation zone, where visibility is often hindered by air produced during ablation. It also provides continuous visualization of the target tumor, enabling precise electrode repositioning to ensure that the ablation zone fully encompasses the tumor. Additionally, tumor viability can be assessed effectively using real-time split-dose FDG-PET [102]. The division of the two FDG doses accounts for the radioactive decay, so that approximately only 10% of the initial dose remains at the final post-ablation scan, which occurs hours after the first injection, thereby minimizing background activity from the original dose. The larger second FDG injection enables precise visualization of the ablation zone and increases sensitivity for detecting hypermetabolic activity from any residual viable tumor after ablation.

The standardized uptake value (SUV) ratio calculated through PET/CT offers a significant predictive capability for LTP in patients with MM of at least 5 mm and negative biopsy results from the AΖ center and margins. This metric has been demonstrated to independently correlate with LTP outcomes in these patients, assisting in the assessment of ablation success and prognosis [104].

CT hepatic arteriography (CTHA) is another available option that provides repeated target tumor depiction before initiation of each ablation cycle [105,106]. CTHA involves placing a catheter selectively into the common or proper hepatic artery to allow repeated injections of small contrast volumes (up to 20 mL) [105]. This approach enhances the visualization of intrahepatic arterial structures and has been shown to significantly improve the precision of targeting CLM [106]. Tumors typically present as a contrast-enhancing ring surrounding a hypoattenuating core, or in cases of LTP after locoregional therapy, as an incomplete hyperattenuating ring [105,106]. At the conclusion of each ablation cycle, repeated contrast injections reveal enhancement within the AZ and highlight any potential residual tumor. The clear visualization of the tumor contours before the procedure and the AZ afterward facilitates image registration and assessment of MM. However, this method has drawbacks, including increased radiation exposure and heightened use of iodine-based contrast agents [105,106].

#### 2.3.5. Biopsy Confirmation

An important limitation of image-guided ablation in comparison to surgery is the lack of pathologic confirmation of complete tumor eradication. Imaging-only assessments presume that the hypo-attenuating area covered by the AZ represents coagulative necrosis of the ablated tumor and surrounding liver parenchyma. Micro-metastases undetectable by imaging adjacent to the target CLM can lead to recurrent tumor growth. In 2000, investigators held a radiologic–pathologic correlation study and demonstrated that the imaging alone was unable to detect 80% of pathologically confirmed residual tumor at the periphery of the AZ [107]. Subsequent studies evaluated the viability of tissue adherent to the electrode by autofluorescence method using glucose-6-phosphate diaphorase staining, morphologic stains, and Ki67 immunohistochemistry [108,109,110]. Presence of viable tumor cells from the ablation zone constituted a strong independent risk factor for LTP. In 2016 the prospective investigation of additional biopsy confirmation for ablation completeness with MM over 5 mm after RFA of CLMs, significantly associated MM size and post-ablation biopsy positivity with LTP [87]. Both MM < 5 mm and biopsy positive for tumor (either from AZ center or margin) were independent predictors of LTP. Recent findings from a prospective, single-group, phase II clinical trial that combined immediate post-thermal ablation tissue and imaging assessments further reinforce the predictive value of tissue confirmation in determining ablation completeness [111].

#### 2.3.6. Patient Selection Bias When Comparing Ablation to Surgery

Controlling selection bias in comparative studies was another important factor contributing to the updated position of percutaneous TA in the treatment algorithm of CLM. Many investigators pointed out that the discrepancies in baseline patient and disease characteristics could not allow reliable comparisons of efficacy outcomes between TA and surgery [64,112,113,114,115]. Surgical candidates presented with a favorable profile for long-term survival and patients with unresectable disease managed with TA were associated by definition with worse prognosis, rendering questionable the reported superiority of surgery for small CLM [64,112,113,114,115]. For these reasons, it was clear there was a need for RCTs comparing ablation to surgery [62,116,117,118].

#### 2.3.7. Patient Eligibility for Ablation and Establishment of a Clinical Risk Score

Establishment of the prognostic factors for oncologic outcomes was another important step in the evolution of ablation [119]. Initial studies demonstrated that tumor size and lymph node positivity serve as independent prognostic factors for worse clinical outcome [63]. In 2011, a pivotal single center study established a 4-scale modified clinical risk score (mCRS) for patients undergoing curative intent ablation for recurrent CLM following surgical resection [9]. Similarly to the surgical clinical risk score (CRS), mCRS contained specific disease characteristics associated with high predictive value for procedural outcomes [9,15,66]. Lymph node-positivity, disease-free survival less than 12 months, ≥1 CLM, and longitudinal diameter of the larger tumor > 3 cm (instead of 5 cm for surgical CRS) constituted the clinical factors and each one counts as one point towards the mCRS; CEA level was not included at that time point due to lack of the corresponding information. High-risk individuals (mCRS ≥ 3) were associated with 3.13 times greater likelihood of LTP in comparison to low-risk patients (mCRS ≤ 2). The reported median overall survival was 21 months for the former group and 35 months for the latter group, respectively [15]. In 2016 the same group reproduced the study for patients with unresectable CLMs or recurrent metastases after metastectomy, with the addition of CEA level > 30 ng/mL (instead of 200 in surgical CRS) as a prognostic factor for LTP and component of mCRS [9]. According to the new 5-scale mCRS, high-risk patients were considered the ones with total score of 4 or 5. Tumor size > 3 cm and ≥1 site of extrahepatic disease (EHD) retained statistical significance after multivariate analysis and presented an inverse relationship with overall survival, while the same relationship was reported for tumor size > 3 cm and MM ≤ 5 mm with LTP-free survival [9].

#### 2.3.8. Treatment Efficacy in Salvage Settings

Thermal ablation has become an established and effective treatment option for recurrence following resection or previous thermal ablation. This approach has demonstrated survival rates comparable to those of salvage surgical resection. Reported overall survival rates for salvage thermal ablation at 1, 3, and 5 years range from 90.1% to 98.9%, 46.2% to 62.6%, and 34.8% to 42.3%, respectively [15,120,121,122].

#### 2.3.9. Procedural Completeness, Treatment Efficacy and Disease Surveillance

Procedural completeness is assessed on the day of ablation. In cases of MM ≤ 5 mm, residual viable tissue on the post-ablation biopsy or FDG uptake when the split dose PET/CT method is used, additional ablation is recommended to achieve technical success [9,92,94,102]. According to guidelines, a prospectively defined time point of 4 to 8 weeks after a technically successful ablation is the recommended time to assess ablation efficacy with a similar process to the intraprocedural MM assessment [72]. The imaging modality recommended for this first evaluation after ablation is ceCT to imitate the intra-ablation assessments [123]. Disease surveillance includes anatomic imaging ceCT and/or MRI with the addition of FDG-PET/CT every 3 to 6 months thereafter, for the first 2 years after the ablation and every 6 months thereafter for a total of 5 years [123].

#### 2.3.10. Guideline Change

The aforementioned evolution and innovations led to an update of the older guidelines that recommended ablation as an alternative option only for unresectable CLM [124,125]. The current guidelines recommend TA as a standalone therapy in highly selected patients with oligometastatic disease with a small number of CLM (less than 3) with maximal longitudinal diameter of 3 cm, as long as all visible disease can be eliminated with appropriate minimal ablation MM ≥ 5 mm (or ≥10 mm in cases of *KRAS* mutant positivity) [41,123,124,125,126].

There was an expected change in the choice of first-line local curative intent treatment for small CLM after the COLLISION trial [127]. This RCT stopped early to meet predefined criteria for non-inferiority in terms of overall survival. The trial demonstrated non-inferiority regarding overall survival and local tumor control, shorter hospitalization time, and fewer complications with TA compared to surgical resection for small-size colorectal liver metastasis up to 3 cm. The assumption that TA should only be used for not optimally resectable colorectal liver metastases changed in the latest update of the guidelines and TA is now considered equivalent to surgical resection for small (≤3 cm) CLM (Figure 4). As a matter of fact, the COLLISION trial reported a better local tumor control rate for TA (by target tumor) when compared to resection, making TA the preferred treatment for small tumors that can be ablated with margins.

## 3. Radioembolization

### 3.1. Trans-Arterial Radioembolization

Trans-arterial radioembolization (TARE) or selective internal radiation therapy (SIRT) is a form of intra-arterial brachytherapy for CLM. Different types of radioactive isotopes presented with favorable outcomes in the settings of prolongation of LTP and reduction in the CEA levels [128,129]. Radioembolization aims to deliver yttrium-90 (^90^Y) selectively to the arterial blood supply of the CLM through the hepatic artery, sparing the surrounding healthy liver parenchyma that is mainly supplied by the portal vein [130,131]. Once the microspheres reach the arterioles, the ^90^Y decays to ^90^Zr, emitting high-energy electrons (β-decay) with a tissue penetration radiation up to 1.1 cm.

Although ^90^Y microspheres utilization for unresectable liver metastases was first reported in 1967, the concept was re-evaluated on human patients in 1992 using glass microspheres (Thera-sphere) as a carrier for the radioactive agent [132,133,134]. According to early reports, the method was associated with up to 70% CEA level reduction, more than 50% decrease in tumor size, local tumor control rate of 85%, and prolongation of the overall survival [133,134,135]. Clinical risk factors associated with worse prognosis were presence of EHD, ≥25% replacement of normal liver parenchyma by the tumor, ECOG score > 2, deprived liver function, time-to-radioembolization interval and history of ≥3 first-line chemotherapeutic regiments [135,136].

Resin microspheres are also used for ^90^Y delivery. Resin-mediated radioembolization was proposed as a well-tolerated locoregional therapeutic option for CLM with possible survival prolongation, particularly in patients with absent EHD [137]. A small phase III RCT of 74 patients with bilobar unresectable metastatic liver disease demonstrated that addition of resin-mediated TARE to hepatic arterial floxuridine (FUDR) chemotherapy statistically significantly increases the liver progression free survival (PFS) and the overall survival rate in the subgroup of patients surviving more than 15 months, without significant procedure-related complications or life quality deprivation [138]. Following this trial, resin-based ^90^Y radioembolization (SIR-spheres) was approved by the Food and Drug Administration (FDA) for treatment of unresectable CLM adjuvant to intra-arterial FUDR. Several large series associated the addition of SIR-spheres to systemic chemotherapy with improved oncologic outcomes and acceptable adverse event profile [139,140,141,142].

An important multicenter phase III clinical trial demonstrated that 5-FU with the addition of a single injection of ^90^Y resin-based TARE to patients with chemorefractory unresectable liver-limited CLM statistically significantly prolongs the LTP interval from 2.1 to 5.5 months [143]. A median overall survival of 12.7 months and a well-tolerated complication rate was reported by a phase I clinical trial in heavily pretreated patients with chemorefractory liver metastases [17].

FOXFIRE, FOXFIRE Global, and SIRFLOX RCTs examined if the addition of resin microsphere to standard of care first-line oxaliplatin-based systemic chemotherapy FOLFOX (5-FU plus leucovorin plus oxaliplatin with elective addition of bevacizumab) would improve survival in chemotherapy naïve patients. A combined analysis of these three trials demonstrated that despite a significant prolongation of the liver-specific progression-free survival there was no impact on progression-free or patient overall survival. Subsequent subgroup analysis in 700 patients with available information regarding the origin of the primary tumor indicated statistically significant prolongation of the overall survival in patients with right-sided origin CRC (22 vs. 17.1 months; *p* = 0.008) [22].

The EPOCH trial (Evaluating TheraSphere in Patients with metastatic colorectal carcinoma Of the liver who have progressed on first-line Chemotherapy) evaluated the efficacy outcomes of adding glass-based ^90^Y radioembolization to standard of care second-line systemic chemotherapy. The study showed statistically significant prolongation of the PFS at patients with both present or absent EHD (9.1 vs. 7 months; *p* < 0.0001) and risk reduction of 31% and 41% for overall and liver disease progression, respectively. There was no difference in overall survival [21].

The aforementioned evidence, along with the technological innovations and equipment development allowing for more selective and precise tumor-specific administration and dosimetry, has attracted the medical community to conduct research and evaluate possible application with curative intent at earlier disease stages with or without the addition of chemotherapy. The current guidelines suggest the utilization of TARE with ^90^Y microspheres can be considered in selected patients with chemotherapy-resistant/refractory disease and with liver-dominant disease [123,126].

### 3.2. Radiation Segmentectomy

Radiation segmentectomy (RS) is the delivery of high ablative radiation dose (at least >190 Gy) to specific liver segments with potential for local cure in carefully selected patients with limited number of CLM not amenable to resection or ablation [144,145]. The available literature on this topic is predominantly focused on patients with hepatocellular cancer. There are three studies reporting outcomes of RS for hepatic metastases (of mixed primary tumor origins), with objective response rates between 44% and 100%, according to RECIST criteria, and LTP-free survival reaching up to 83% [146,147,148]. A single center retrospective study evaluating 36 patients with 57 CLM treated with RS demonstrated that mean tumor absorbed dose ≥ 400 Gy and mean dose absorbed at 5 mm margins around the tumor ≥ 350 Gy are independent predictive factors of prolonged LTP-free survival [149]. The same study showed that stereotactic coverage of at least 95% of the tumor with a dose over 350 Gy was associated with sustained, long-term local tumor control. This study pointed out the importance of post-RS dosimetry on oncologic outcomes, an area of much needed and continued improvement [149]. Current guidelines suggest that the approach can be considered for small tumors that cannot be resected or ablated [123].

### 3.3. Radiation Lobectomy

Radiation lobectomy is the infusion of high dose ^90^Y within the tumor involved hepatic lobe, in order to achieve local disease control while simultaneously inducing functional future liver remnant (FLR) hypertrophy in potentially or borderline resectable patients. The method was first introduced in 2009 in a small series of hepatocellular carcinoma and cholangiocarcinoma cases [150]. The results are promising in terms of liver hypertrophy and local disease control [151,152,153]. A small single center series of unresectable CLM showed superiority of portal vein embolization (PVE) over radiation lobectomy in terms of the degree of FLR hypertrophy, probably due to the slower rate of contralateral hypertrophy induced by TARE when compared to PVE [154]. In another series of 83 patients with right unilobar extensive hepatic metastases, including eight patients with CLM, the median functional liver remnant hypertrophy reached 45% at nine months [155]. Another small series of five patients with CLM combining portal vein and hepatic vein embolization, as well as radiation lobectomy, resulted in significant liver remnant hypertrophy with concomitant local disease control [156]. According to the current guidelines, radiation lobectomy can be considered instead of PVE when hepatic metastatic disease is not optimally resectable based on insufficient remnant liver volume or when there is borderline resectable disease that would benefit from tumor downsizing and remnant hypertrophy [123,126].

### 3.4. ^90^Y TARE Planning and Predictors of Oncologic Outcomes

Proper treatment planning is required to achieve selective and accurate ^90^Y delivery, in an effort to maximize the radiation dose to the tumor while sparing normal liver parenchyma. Numerous studies have demonstrated the importance of personalized dosimetry during procedure planning. Post-treatment dosimetry can identify patients who will achieve prolonged response. Ongoing studies are evaluating the outcomes of synergy with other locoregional and systemic treatments, as well as the role of other radiopharmaceuticals for intra-arterial delivery.

#### 3.4.1. Mapping Arteriography

Treatment planning with mapping arteriography is required prior to TARE. In this procedure, the hepatic arterial anatomy is evaluated with arteriography. Cone beam CT or CT angiography (in combined fluoro/CT-angio suites) is often necessary to delineate the perfused liver volume and ensure coverage of the target tumor(s), as CLMs are typically hypovascular. Aliquots of ^99m^Technetium macroaggregated albumin (^99m^Tc-MAA) are injected into the arterial branches where ^90^Y administration is planned/intended [151]. Patients are subsequently transferred to the nuclear medicine department for Single Photon Emission Computed Tomography-Computed Tomography (SPECT/CT) imaging. This scan shows the distribution of the radiotracer and allows calculation of the T:N (tumor to normal liver) ratio and lung shunt fraction (LSF). Moreover, extrahepatic radiotracer distribution can be identified and addressed with embolization at the time of the treatment or usage of antireflux catheters [20,152]. The information acquired from the mapping arteriogram is required for calculation of the ^90^Y activity that will be ordered and delivered during treatment. In cases of LSF ≥ 20%, due to increased risk of pneumonitis, reduction in dose or temporary occlusion of hepatic veins to decrease the shunt and allow safe ^90^Y administration is required [153]. The absolute threshold of total lung exposure to radiation for both glass- and resin-based TARE is 30 Gy [20]. Whenever a temporary vessel occlusion is required for redistribution of flow to the tumor and prevention of non-target ^90^Y tissue deposition, embolization is recommended at the same session with TARE procedure day and not during the mapping procedure to avoid recanalization or induction of collateral vasculature that can develop between the mapping and the day of ^90^Y administration [157].

#### 3.4.2. Dosimetry

The literature supports a strong association between ^90^Y dose and treatment response for both resin and glass microspheres [158,159,160,161]. In a small series of 45 patients with CLM treated with SIR-spheres, a median tumor dose of 100 Gy prolonged the overall survival and improved objective local tumor response [162].

Evidence from RS studies also supports the dose–response relationship. In a small series of 36 patients with hepatic metastases (11 CLM; 31%) undergoing RS with target radiation dose > 200 Gy resulted in 92% local disease control rate; 28% of the patients ultimately progressed at a median follow-up of 12 months [148]. Another small case series of 10 hepatic metastases including 7 CLM indicated that a mean tumor absorbed dose of 251.7 Gy, achieving a local disease control rate of 85.8% [147]. According to the previously mentioned single center RS study including exclusively patients with CLM, a mean tumor absorbed dose ≥400 Gy and a 5 mm margin around the tumor mean absorbed dose ≥350 Gy; the LTP rate was 16.7% in contrast to 86% LTP rate of the cases with lower doses (Figure 5) [149].

There are different methods used to calculate the ^90^Y administration dose. The initially reported method of dosimetry was utilizing the Body Surface Area (BSA) model. This technique, commonly used for SIR-spheres, calculates the prescribed dose using the BSA and the tumor burden within the target liver parenchyma [163,164]. In an effort to increase the ability of operators to provide personalized dosimetry based on tumor characteristics and therapeutic intent, a small patient trial of 15 patients using Tc99-MAA for LSF calculation, CT to calculate liver volume, and post-TARE bremsstrahlung scan for actual ^90^Y delivery location confirmation, generating the Medical Internal Radiation Dose (MIRD) model [165]. MIRD utilization is recommended by Thera-sphere manufacturers and was also used in the EPOCH trial. Main limitation derives from the fact that it is a single-compartment model, which contains both the target tumor and surrounding healthy parenchyma (Figure 6). Dose volume histogram (DVH) curves that display the dose distribution within the different generated compartments and the included structures and tumor (Figure 7).

The most advanced method available is partition model dosimetry. It depends on Tc-99m-MMA SPECT/CT and utilizes multiple different compartments for the tumor of interest, the non-tumor liver parenchyma and the lung. This model allows activity administration doses to be separately calculated and used as thresholds for each individual compartment. Main limitations of this method are the different biodistribution between Tc-99m-MAA and ^90^Y, variability in catheter position and actual dose administration between the mapping and radioembolization procedure and the assumption that the dose is uniformly distributed within each compartment [166,167]. Further evolution with subsequent precision in dosimetry will optimize oncologic outcomes and minimize adverse events.

#### 3.4.3. Pre-TARE Predictors of Treatment Response for Patient Selection Optimization, Hepatic Arterial Infusion Pump Synergy and Other Isotopes

Defining factors that affect the clinical outcome is crucial for improving patient selection and consequently oncologic outcomes after TARE. CEA ≥ 150 ng/mL, transaminase ≥ 2.5 × upper normal limit, sum of 2 largest liver diameters on CT ≥ 10 cm and history of liver surgery prior to TARE albumin level, tumor differentiation level and number of sites of EHD were documented as independent prognosticators of overall survival [17,168,169,170,171]. Pre-TARE imaging characteristics have also been assessed. SUV maximum tumor uptake has been statistically significant predictor of LTP [169]. Tumor arterial enhancement fraction, (defined as the arterial phase enhancement divided by the portal phase enhancement at a pre-treatment triphasic liver scan) has been supported to predict the metabolic response on post-^90^Y PET/CT [130]. Additional imaging biomarkers evaluating treatment response have also been suggested as predictors of LTP-free and overall survival for PET/CT and MRI modalities [172,173,174].

Another subject of interest for TARE in CLM management is the synergy with hepatic arterial infusion pump (HAIP). Literature supports clinical outcomes improvements from the addition of radioembolization to HAIP chemotherapy [17,138,175]. Investigators assessed the value of the addition of a single dose of ^90^Y through the HAIP in patients with extensive liver metastases [137]. After a time-interval of four weeks selective arterial 5-FU chemotherapy was re-initiated. The results supported improvement of survival times, especially in patients with absence of extra-hepatic disease [137]. Another retrospective review evaluating factors that affect the efficacy outcome of radioembolization in heavily pretreated patients reported prolongation of the overall survival in study participants who received additional HAIP-mediated chemotherapy after TARE [17]. In a prospective phase I clinical trial assessing TARE efficacy and safety outcomes in heavily pretreated patients who experienced local tumor progression, additional HAIP chemotherapy after radioembolization resulted in 50% reduction in CEA levels [175]. Current evidence does not prohibit re-initiation of pump use upon progression after TARE provided that liver function enzymes (LFTs) and bilirubin levels remain acceptable [17,138,175]. It is reasonable to investigate ways to optimize the combination of these modalities in subsequent work. There is a recently announced RCT that intends to evaluate the combination of TARE and HAIP for unresectable HCC (NCT06867432). Lastly, the bibliography almost exclusively contains studies evaluating the effect of ^90^Y radioembolization with very sparse data for other isotopes [176,177]. Evaluation of the efficacy of other radio-active microspheres can greatly contribute to the optimization of clinical outcomes of TARE on metastatic colorectal cancer to the liver.

#### 3.4.4. Treatment Response Assessment and Disease Surveillance

Main components of the expected response to TARE on imaging is target tumor devascularization with consequent hypo-attenuation, necrosis, and reduction in size [178,179]. At early follow-up pseudo-progression manifests as an increase in tumor diameter at all planes secondary to intra-tumoral edema and hemorrhage secondary to necrosis. Therefore, treatment efficacy is evaluated at ceCT or ceMRI scan 8 to 12 weeks after TARE [178,179].

Metabolic response by FDG-PET/CT usually precedes imaging response by ceCT and ceMRI. Metabolic response of CLM can be detected as soon as 4 to 6 weeks after treatment [20,180]. Demonstrating response early can be crucial for patients requiring bilobar radioembolization. Confirming disease response to ^90^Y justifies utilization of the treatment strategy to the contralateral hepatic lobe (Figure 8). On the contrary, in cases with lack of metabolic response or progression within the treated hepatic area, alternative treatments can be proposed. Continuous disease monitoring is recommended thereafter every 2–4 months for at least 1 year [172,181].

## 4. Chemoembolization

Trans-arterial chemoembolization (TACE) includes selective catheterization of the hepatic arterial blood supply to the tumor and administration of embolic material combined with chemotherapeutic agents. Vessel occlusion can be either temporary or permanent, depending on the embolic agent used. The method was first described in 1981 [182]. In 1993, TACE with ethiodized oil and doxorubicin for 46 patients with resectable primary colorectal cancer and liver-limited metastatic disease demonstrated increase in the overall survival and PFS; 6 patients failed to respond [183].

### 4.1. Mitomycin-C

Early reports combining mitomycin-C and either 5-FU or doxorubicin and cisplatin were associated with overall survival and LTP-free survival prolongation [184,185]. Patient performance status, liver-limited disease and low levels of alkaline phosphatase and lactate dehydrogenase were suggested as independent prognosticators of overall survival [184]. Median overall survival was significantly improved when chemoembolization was performed after first- or second-line systemic chemotherapy, in comparison to cohorts where this treatment was utilized after third- to fifth-line chemotherapy, although this likely represents lead time bias [186]. The chemotherapeutic agent was evaluated in different drug combinations with objective radiologic response ranging between 14.7% and 77% and the median overall survival between 8.6 and 14 months [186,187,188]. The evidence supported Mitomycin-based TACE to be well tolerated by patients, with the exception of a phase II clinical trial assessing mitomycin C, with Lipiodol demonstrating no clinical benefit and significant liver toxicity [189]. A recently published single center retrospective study evaluated selective intra-arterial, sequential monthly Mitomycin infusions in combination to bi-weekly systemic irinotecan in patients that progressed after HAIP chemotherapy with floxuridine and at least two prior lines of systemic chemotherapy [190]. The median PFS, liver PFS and overall survival were 4.1, 5.5, and 9.6 months, respectively. Disease progression in 56.5% of the cases and toxicities associated with the DNA alkylating agent in 39.1% of the cases resulted in discontinuation of the infusion protocol. Hepatobiliary complications, occlusion/thrombosis of the target artery for the infusion, and severe pneumonitis lead to cessation of infusion in 20%, 15%, and 4% of the cases, respectively. Toxicities were mainly observed in patients receiving ≥4 Mitomycin infusions [190].

### 4.2. DEBIRI

A phase II clinical trial introduced the idea of irinotecan drug-eluting beads (DEB) trans-arterial administration for CLM management, resulting in significant reduction in CEA levels and CLM contrast enhancement [191,192] 4 weeks post-treatment. The treatment was well-tolerated, with right upper quadrant pain reported by all of the patients. Other phase II trials and single center studies also showed acceptable dose-dependent adverse event rate and superiority of DEBIRI over conventional TACE [193,194,195,196,197,198,199,200,201].

### 4.3. Comparison of TACE to Other Local Treatments

Comparison of TARE and TACE for the management of CLM in the salvage setting concluded in similar efficacy results and complication rates [202]. RCT comparing standard of care systemic FOLFIRI administration to DEBIRI-TACE resulted in an increase in the overall survival of the DEBIRI arm [27]. More specifically, the overall survival was increased in cases at which DEBIRI was initiated after first- or second-line chemotherapy, reaching 11 to 12 months, in contrast to patients receiving DEBIRI after progression on third line of systemic chemotherapeutic regiment achieving survival of 6 months [27]. DEBIRI, in comparison to systemic chemotherapy (FOLFIRI), was associated with prolonged median survival (22 vs. 15 months), greater likelihood of objective tumor response in the liver (68.6% vs. 20%), prolonged time to EHD progression (13 and 9 months), and improved physical functioning during active treatment. DEBIRI was also evaluated in combination to first-line systemic chemotherapy through comparison of DEBIRI in combination with FOLFOX versus FOLFOX with or without the addition of bevacizumab. Even though patients of DEBIRI FOLFOX arm had a worse ECOG score and greater proportion of patients with EHD, this group was associated with better response rates, higher percentage of patients downsizing to resection, and improvement in the median PFS [203].

Combination of TACE and MWA versus MWA monotherapy demonstrated superiority of the former in the settings of hepatic PFS and overall survival in salvage settings [204]. The CIREL observational study recently published results for irinotecan-based TACE as salvage or consolidation/post-inductive treatment and correlated the method with long median overall survival when utilized in salvage settings and promising hepatic PFS when utilized along with systemic chemotherapy or TA [205]. The authors proposed further investigation of the utility of TACE and possible incorporation in earlier stages in the management of CLMs.

Currently, DEBIRI is recommended as salvage treatment in combination to second-line chemotherapeutic regiments aiming to achieve local disease control and potentially downsize a CLM to a resectable or ablatable stage [123,126]. The principles of follow-up imaging recommendations are the same as with TARE [123,126].

## 5. Conclusions

There has been continuous evolution in the role of interventional oncology for the treatment of CLM, both in the locally curative and salvage settings. These minimally invasive treatments can be used alone or in combination with other systemic or locoregional therapies to provide disease control. Thermal ablation is now recognized as a standalone curative intent treatment option, considered equivalent to surgical resection for metastases with a maximal diameter of 3 cm. Radiation segmentectomy is also employed with potentially curative intent in carefully selected patients with a limited number of CLM not amenable to resection or ablation, and advancements in dosimetry may allow ^90^Y trans-arterial radioembolization to be incorporated earlier in management algorithms for colorectal liver metastases. Furthermore, emphasis should be placed on evaluating emerging modalities such as histotripsy, with thorough investigation of their safety and efficacy profiles in order to define the patient subgroups most suitable for these novel treatments. Ongoing clinical trials aim to evaluate the role of each treatment modality in the treatment paradigm.

## Figures and Tables

**Figure 1 biomedicines-13-02182-f001:**
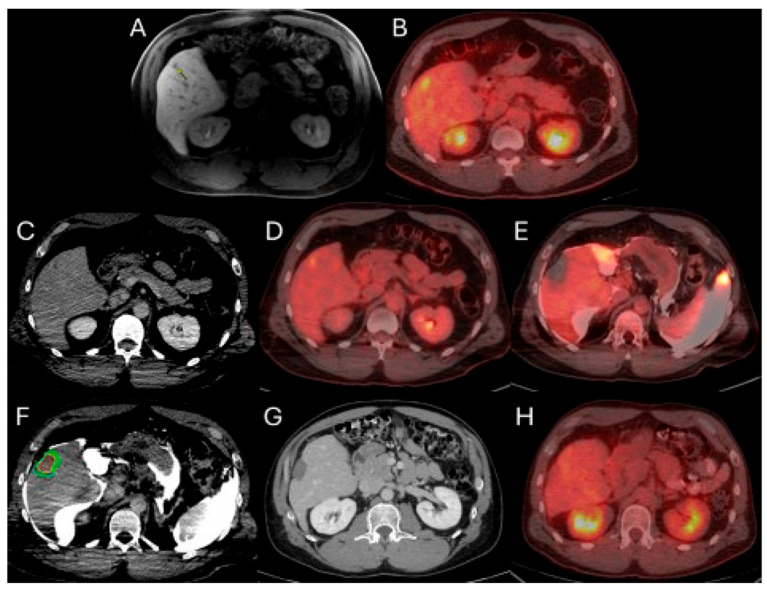
56-year-old male with history of colorectal cancer under chemotherapy treatment presents with new segment 5 liver metastasis on T1 Magnetic Resonance Imaging with fat suppression (**A**) and fluorodeoxyglucose Positron Emission Tomography and Computer Tomography (FDG-PET/CT) (**B**) and undergoes microwave ablation (MWA). The target tumor is visible on the contrast enhanced Computer Tomography (ceCT) on ablation day imaging (**C**). Administration of the first FDG dose according to the split-dose PET/CT protocol (4 mCi), the FDG-avid tumor is clearly visualized (**D**). Hydrodissection is utilized to protect the adjacent ascending colon. After administration of the second FDG dose (8 mCi), the photopenic ablation zone indicates absence of residual metabolic activity (**E**). Ablation technical success is confirmed on post-ablation ceCT with utilization of MIM DEV software version 3.3.7. Red circle represents the tumor contours, yellow circle and green circle represent the 5 and 10 mm (mm) margins around the target tumor, respectively. The blue circle represents the contours of the ablation zone that uniformly covers the tumor with 5 mm margins, confirming technical success of thermal ablation (**F**). Follow-up anatomic and metabolic imaging at 2 years demonstrates involution of the ablation zone (**G**) with sustained local disease control (**H**).

**Figure 2 biomedicines-13-02182-f002:**
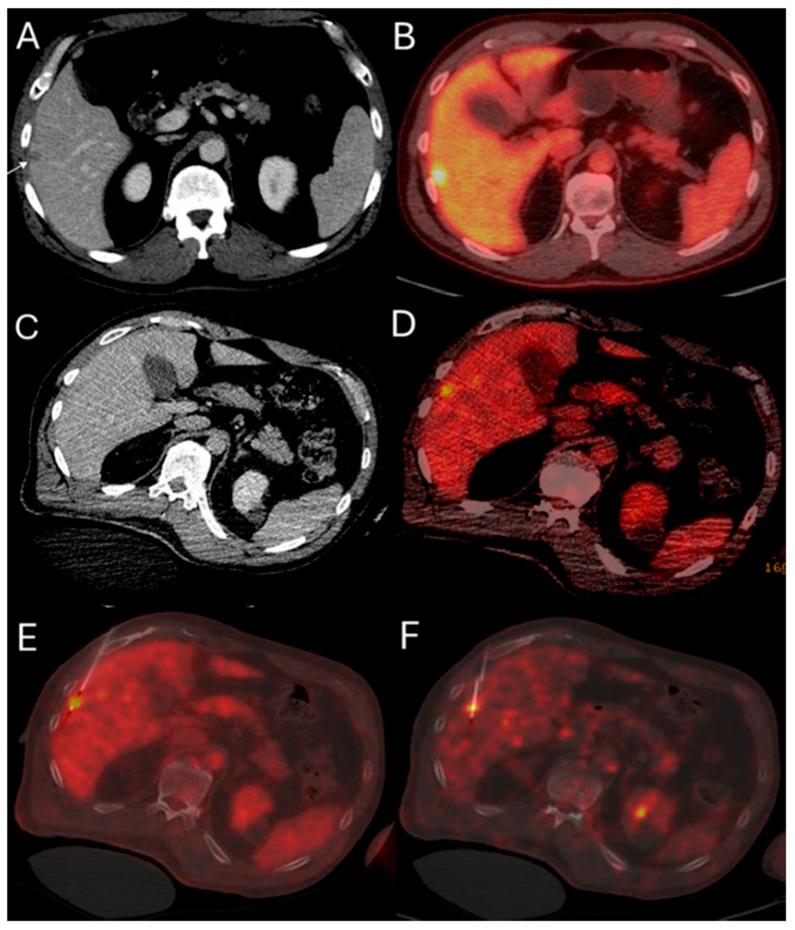
51-year-old male with history of colorectal cancer undergoes MWA for new subcapsular liver metastasis in hepatic segment 5. Pre-ablation ceCT ((**A**) arrowhead) and FDG-PET/CT (**B**) demonstrate the tumor. The intraprocedural ceCT barely identifies the target tumor (**C**). Utilization of split-dose FDG PET/CT allows visualization, targeting, and monitoring of the FDG-avid tumor (**D**) with short acquisition 1 min breath-hold real-time scans allowing for accurate targeting for biopsy (**E**) and microwave ablation (**F**).

**Figure 3 biomedicines-13-02182-f003:**
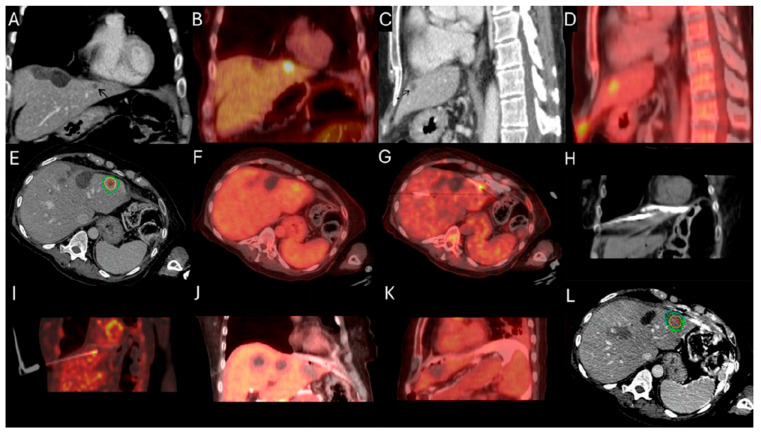
63-year-old female with metastatic colorectal cancer liver dominant disease, previously treated with chemotherapy, hepatectomy, and intraoperative thermal ablations at segment 8, undergoes MWA for new colorectal liver metastasis (CLM) in segment 2. Pre-procedure ceCT ((**A**,**C**), black arrowhead) and FDG-PET (**B**,**D**) demonstrate the subdiaphragmatic metastasis in close proximity to the coronary sinus and the pericardium. MIM DEV version 3.3.7 semi-automatically generated target tumor and margin contours: using the intra-procedure pre-ablation ceCT, red circle represents the tumor contours, yellow and green circles represent the 5 and 10 mm (mm) margins around the target tumor, respectively (**E**). Administration of the first dose (4mCi) of FDG according to split-dose protocol provides clear visualization of the FDG-avid tumor (**F**). Extensive hydrodissection creates a buffer between the pericardium and the edge of the liver (**G**,**H**). Short acquisition 1 min breath-hold real-time FDG PET/CT scans confirm accurate placement of the microwave ablation electrode during hydrodissection (**I**). Second injection of 8mCi FDG confirms absence of residual metabolic activity (**J**,**K**). Post-ablation ceCT demonstrates the ablation zone (blue circle) uniformly covering in 3-dimension (3D) the tumor with 5 mm margins, confirming ablation completeness (**L**).

**Figure 4 biomedicines-13-02182-f004:**
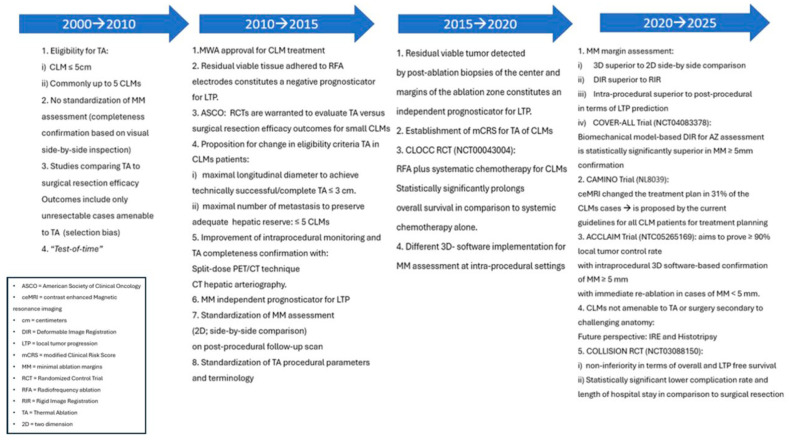
Thermal ablation evolution.

**Figure 5 biomedicines-13-02182-f005:**
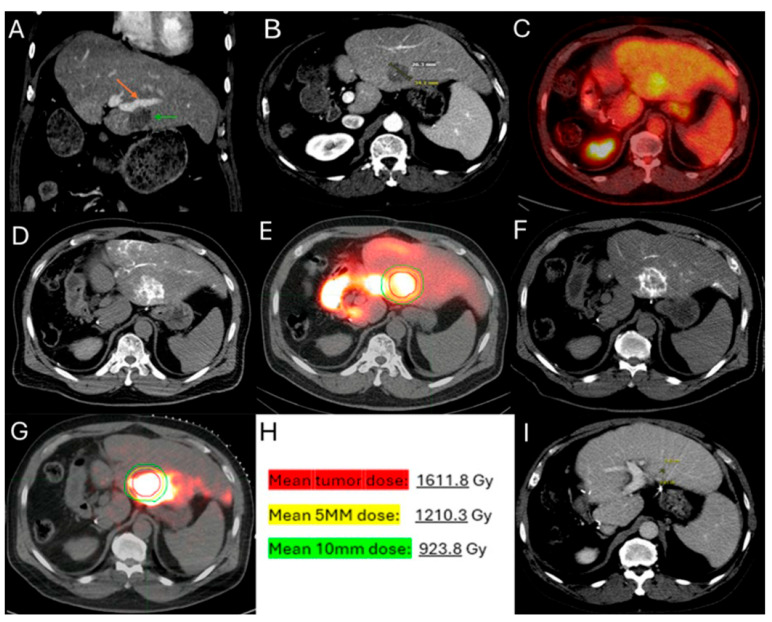
60-year-old male with history of Neuroblastoma Rat Sarcoma (*N-RAS)* mutant rectal adenocarcinoma with bilobar liver metastases and prior systemic chemotherapy, right hepatectomy, and left metastasectomies, presents with new metastasis in close proximity to the left portal vein deemed high risk for thermal ablation (**A**–**C**). After multidisciplinary review, radiation segmentectomy was recommended as the appropriate therapy to minimize the risk for injury of the nearby portal vein ((**A**) orange arrowhead indicates the left branch of portal vein, green arrowhead indicates the target tumor). During mapping, CT arteriography demonstrates optimal tumor coverage (**D**) confirmed by Technetium-99m Macroaggregated Albumin (Tc-99m-MMA SPECT/CT) (**E**). CT arteriography during TARE (**F**) and post-TARE bremsstrahlung scan (**G**) demonstrates optimal target tumor coverage (**E**,**G**): the red circle represents the tumor contours, while the yellow the 5 mm and the green the 10 mm margin areas around the target tumor. The mean tumor absorbed dose is 1611.8 Gray (Gy), the mean 5 mm absorbed dose 1210.3 Gy, and the mean 10 mm margin absorbed dose is 923.8 Gy according to dosimetry workflow of MIM DEV version 3.3.7. (**H**). Follow-up ceCT at 2 years indicates sustained long-term tumor control (**I**).

**Figure 6 biomedicines-13-02182-f006:**
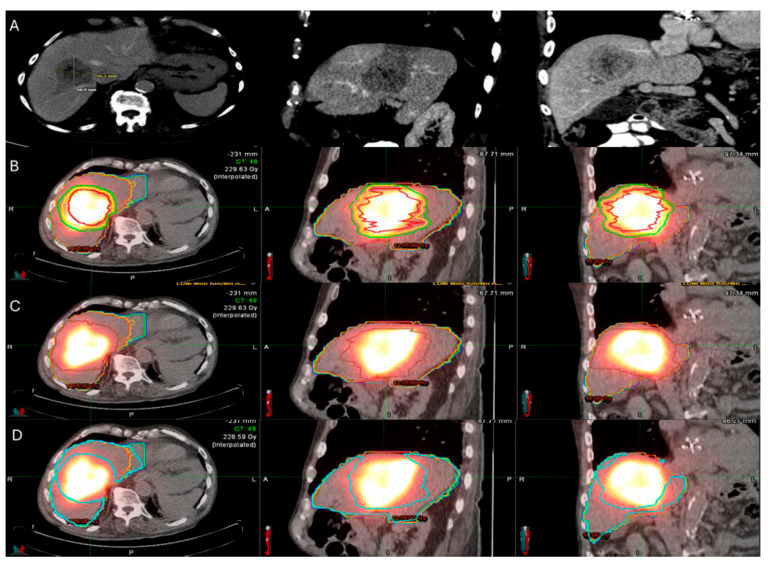
86-year-old with synchronous relatively large metastasis in the right lobe on pre-treatment ceCT (**A**) undergoes radiation segmentectomy. Post-TARE Medical Internal Radiation Dose (MIRD) dosimetry is demonstrated according to the workflow using MIM DEV software version 3.3.7 using the post-TARE bremsstrahlung scan (**B**–**D**). The contours of the treated tumor are illustrated by the red line; yellow circle represents the 5 mm margins, and the green circle represents the 10 mm margins around the target tumor (**B**). The area defined by the brown circular line represents the single compartment used by the MIRD method that contains the target tumor (**C**). The area enclosed by the blue line represents the non-tumor liver parenchyma (**D**).

**Figure 7 biomedicines-13-02182-f007:**
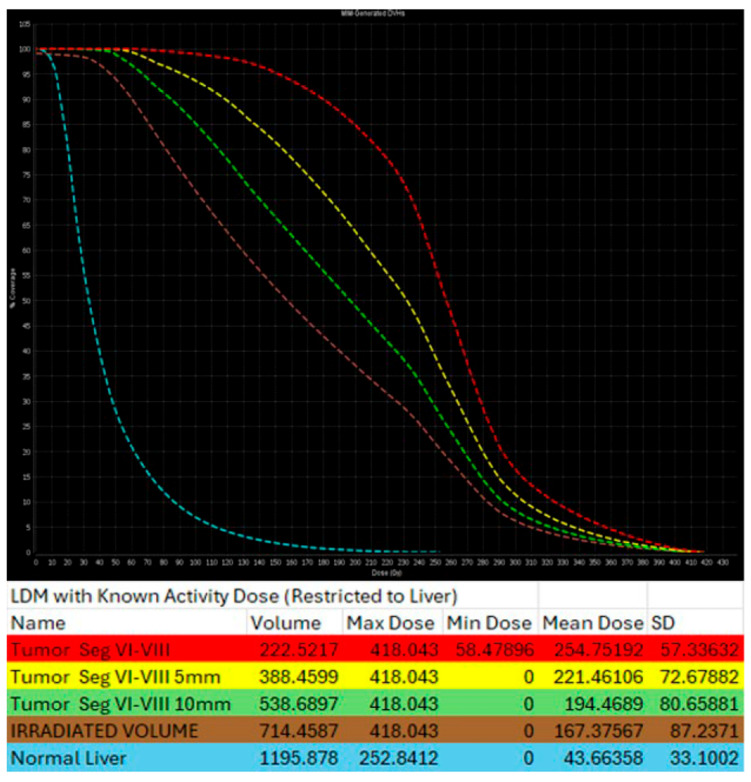
Same patient of Figure 6. Post-TARE MIRD dosimetry Dose Volume Histogram (DVH) curve (% volume coverage ~ actual dose delivered) with utilization of MIM software version 3.3.7. DVH demonstrates dose distribution at the target tumor (red color), at the 5 mm margins (yellow color) and 10 mm margins (green color) around the tumor, at the liver single compartment created by the software that contains the target tumor (brown color), and at the non-tumor liver parenchyma (blue color).

**Figure 8 biomedicines-13-02182-f008:**
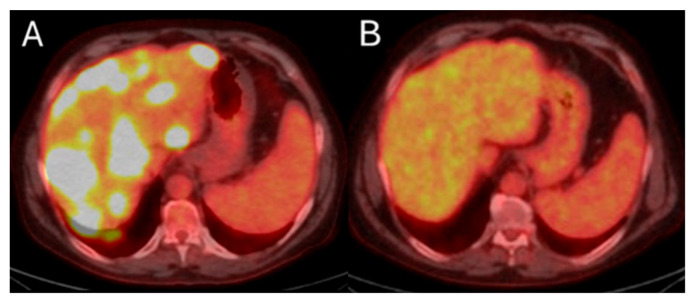
55-year-old male with history of colorectal adenocarcinoma currently on systemic chemotherapy undergoes trans-arterial radioembolization with Ytrium-99 microspheres for multifocal liver dominant metastatic progressive chemorefractory disease in salvage settings. Pre-procedure FDG-PET/CT indicates diffuse bilobar disease (**A**). The eight months post-initial TARE demonstrates sustained disease control while the patient is under chemotherapeutic coverage (**B**).

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
