# Peer review of "Interventional Oncology for Colorectal Liver Metastases: From Local Cure to Salvage Therapy"

_biomedicines, 2025, doi:10.3390/biomedicines13092182_

Round 1
Reviewer 1 Report
Comments and Suggestions for Authors
The extent of coverage for each interventional modality varies considerably. For instance, thermal ablation techniques regarding evolution, target size, location, and margin assessment are extensively analyzed. In contrast, other modalities such as Irreversible Electroporation (IRE) and Histotripsy are examined in less detail, primarily emphasizing mechanism and preliminary trials. Similarly, although intra-arterial therapies (TARE, TACE) are included, the discourse on their advancement, especially about patient selection criteria beyond the broad designation of "unresectable," appears less detailed than ablation.
Question: In light of the assertion of a "comprehensive review," what explicit criteria were employed to ascertain the degree of detail offered for each interventional modality? Was there a methodical approach to selecting and prioritizing the aspects addressed for each treatment? Was a comprehensive literature search conducted for newer or less established procedures to ensure the inclusion and critical evaluation of all pertinent recent advancements and clinical trials, beyond those cited?
The study consistently emphasizes the benefits of interventional procedures, especially thermal ablation, compared to surgical resection, noting factors such as reduced morbidity, abbreviated hospital stays, and diminished expenses. Although these observations are pertinent, the text might improve with a more equitable examination of prospective long-term oncologic outcomes (e.g., survival rates, recurrence patterns) when contrasting interventional therapy with surgical options, particularly in resectable instances. The discourse surrounding the COLLISION trial (Ref 126) predominantly emphasizes the non-inferiority of ablation; however, it is crucial to address the specific limitations or subtleties of this non-inferiority conclusion, particularly in light of the trial's premature cessation.
Question: Although the COLLISION trial is referenced as evidence for non-inferiority, what are the precise long-term oncologic outcomes (e.g., 5-year overall survival, disease-free survival) that substantiate thermal ablation's equivalency to surgery beyond the initial 2-year data? Are there particular patient subgroups or tumour characteristics for whom surgery may provide a greater long-term oncologic advantage, and if so, is this sufficiently highlighted? he assertion that "the COLLISION trial reported a superior local tumour control rate for TA (by target tumour) compared to resection, establishing TA as the preferred treatment for small tumours amenable to ablation with margins" (Page 10) seems to be an excessively robust conclusion, particularly in light of the non-inferiority design. uld the authors clarify how a "better local tumour control rate" directly informs a preferable therapy decision within the framework of a non-inferiority trial, particularly considering the potential impact of factors like as patient selection or follow-up duration on this interpretation?
The evaluation delineates the indications for each treatment method in general terms (e.g., "small CLM suitable for ablation with sufficient margins," "more extensive disease that is not suitable for resection or ablation"). The intricacies of patient selection within these categories, particularly with multidisciplinary team (MDT) talks, patient performance status, comorbidities, and previous systemic medications, warrant more elaboration.
Question: For each interventional modality addressed, in addition to tumour size and resectability, what are the essential patient-specific factors (e.g., ECOG performance status, liver function, systemic chemotherapy regimen, genetic mutations such as KRAS/BRAF beyond their reference for margin evaluation) that influence the decision-making process in a practical multidisciplinary team setting? Publication indicates that "current guidelines recommend TA as a standalone therapy (table 1)," although Table 1 is absent from the supplied PDF. Evaluation of the guideline revisions is unfeasible without Table 1. Kindly furnish Table 1 and specify its source (e.g., NCCN, ESMO, or other national/international guidelines) along with the level of recommendation for each situation.
The manuscript briefly discusses the synergy between percutaneous TA and systemic chemotherapy (e.g., CLOCC trial). It similarly references TARE in conjunction with HAIP treatment. A more systematic and analytical examination of the appropriate sequencing of interventional therapies alongside systemic treatments (e.g., neo-adjuvant versus adjuvant, concurrent versus sequential) and the data for these combinations would improve the review.
Question: What are the principal factors and problems in developing clinical trials to explore the appropriate sequencing and combination of systemic chemotherapy with interventional oncology treatments for colorectal liver metastases (CLM)? What are the deficiencies in evidence concerning multi-modality techniques, and which sorts of studies are most essential to rectify these deficiencies?
The review examines the advancement of imaging modalities for margin evaluation (2D versus 3D, intraprocedural versus post-ablation) and the function of PET/CT and CTHA in monitoring. It also references biomarkers such as CEA and KRAS status. A more unified section on incorporating modern imaging and molecular indicators into treatment planning, response assessment, and recurrence surveillance, rather than isolated references, would be advantageous.
Question: How do contemporary imaging modalities (e.g., improved MRI sequences, quantitative PET metrics, AI-enhanced image processing) presently or potentially enhance patient selection for targeted interventional therapies? Aside from KRAS, are there further developing molecular biomarkers (e.g., circulating tumour DNA, gene expression profiles) that can forecast responses to specific interventional therapies or inform their use, and if so, how could they be incorporated into clinical practice for colorectal liver metastases (CLM)?
The abstract and discussion mention "salvage therapy." A comprehensive examination of the long-term results for patients undergoing interventional therapies in a salvage context is necessary, notably regarding overall survival, quality of life, and the management of later recurrences, despite the initial benefits of these treatments being outlined.
Question: What specific problems arise in defining and assessing long-term outcomes (e.g., following lines of therapy, cumulative toxicity, overall survival) for patients receiving interventional oncology procedures as salvage therapy for colorectal liver metastases (CLM) in severely pre-treated populations? What is the standard time and method of follow-up for patients undergoing salvage interventional treatments, and how does this contrast with those receiving curative purpose therapies?
The author could use the following paper (if applicable):
Identification of Novel Molecular Panel as Potential Biomarkers of PAN-Gastrointestinal Cancer Screening: Bioinformatics and Experimental Analysis
Sparassis latifolia and exercise training as complementary medicine mitigated the 5-fluorouracil potent side effects in mice with colorectal cancer.
Author Response
Response to the Editor and Reviewers
Biomedicines
RESPONSE TO COMMENTS
Ref.: Manuscript ID: biomedicines-3812108
Interventional Oncology for Colorectal Liver Metastases: From Local Cure to Salvage Therapy
Dear Dr. Manish Tripathi and Dr. Puneet Vij,
We sincerely thank you and the reviewers for your thoughtful and constructive evaluation of our manuscript. We greatly appreciate the insightful comments and suggestions, which have been highly valuable in improving the quality and clarity of our work. We have carefully addressed each point in a detailed, point‑by‑point response below:
REVIEWER 1#
The extent of coverage for each interventional modality varies considerably. For instance, thermal ablation techniques regarding evolution, target size, location, and margin assessment are extensively analyzed. In contrast, other modalities such as Irreversible Electroporation (IRE) and Histotripsy are examined in less detail, primarily emphasizing mechanism and preliminary trials. Similarly, although intra-arterial therapies (TARE, TACE) are included, the discourse on their advancement, especially about patient selection criteria beyond the broad designation of "unresectable," appears less detailed than ablation.
Question: In light of the assertion of a "comprehensive review," what explicit criteria were employed to ascertain the degree of detail offered for each interventional modality? Was there a methodical approach to selecting and prioritizing the aspects addressed for each treatment? Was a comprehensive literature search conducted for newer or less established procedures to ensure the inclusion and critical evaluation of all pertinent recent advancements and clinical trials, beyond those cited?
We sincerely thank the reviewer for their thoughtful and constructive comment. In this review article, our primary aim was to provide readers with a comprehensive overview of the evolution of each interventional oncology modality currently employed in the treatment of colorectal liver metastases. To achieve this, we conducted an extensive search of Cochrane Library, Embase, Scopus, and PubMed, focusing on original studies, systematic reviews, and meta-analyses that offered pivotal insights into the development and clinical application of these modalities.
For clarity and readability, we structured the manuscript into distinct subunits dedicated to each therapeutic approach, thereby allowing readers to follow the stepwise technical advancements that have shaped current clinical practice. For every subunit, we repeated the search strategy without year restrictions, covering publications up to June 2025. This method ensured that all relevant studies were identified and that each modality was discussed in sufficient depth, as also reflected in the extensive citation list provided throughout the manuscript.
Given that the special issue focuses on major advancements in colorectal liver metastasis management over the past two decades, we considered it important to present studies in a comprehensive and chronological manner, highlighting the progression of techniques and evidence supporting their adoption in clinical practice. The design of this review is narrative and not systematic. Because our literature search methodology was consistently applied to each treatment modality and within each relevant subunit, we believe that a separate methodological unit may detract from clarity and risk confusing the reader. To avoid this frustration and according to your recommendation, we rephrased the introduction (section 1) as follows according to your indications:
“This comprehensive narrative review describes the evolution of interventional oncology treatments for CLM and examines the appropriate indications for each treatment modality.”
We appreciate the reviewer’s observation regarding the varying depth of coverage of the different interventional modalities in the manuscript. This variation reflects the fact that the pace of technological development and the availability of high-quality evidence have differed substantially among modalities over the past two decades. In order to remain consistent with the existing body of evidence, the extent of our discussion for each treatment is proportional to the degree of evaluation and emphasis seen in the NCCN guidelines.
For example, thermal ablation techniques have been extensively investigated and validated, resulting in a more detailed discussion. In contrast, modalities such as irreversible electroporation are supported by a more limited body of evidence, while histotripsy has only very recently entered clinical practice for colorectal liver metastasis management. Consequently, the available data for these newer techniques are sparse compared with more established treatments. Nonetheless, to the best of our knowledge, this review compiles and synthesizes the most comprehensive collection of evidence regarding histotripsy for colorectal liver metastases currently available in the literature.
We appreciate the reviewer’s perspective, and we hope that this explanation clarifies our rationale for the chosen structure.
The study consistently emphasizes the benefits of interventional procedures, especially thermal ablation, compared to surgical resection, noting factors such as reduced morbidity, abbreviated hospital stays, and diminished expenses. Although these observations are pertinent, the text might improve with a more equitable examination of prospective long-term oncologic outcomes (e.g., survival rates, recurrence patterns) when contrasting interventional therapy with surgical options, particularly in resectable instances. The discourse surrounding the COLLISION trial (Ref 126) predominantly emphasizes the non-inferiority of ablation; however, it is crucial to address the specific limitations or subtleties of this non-inferiority conclusion, particularly in light of the trial's premature cessation.
Question: Although the COLLISION trial is referenced as evidence for non-inferiority, what are the precise long-term oncologic outcomes (e.g., 5-year overall survival, disease-free survival) that substantiate thermal ablation's equivalency to surgery beyond the initial 2-year data? Are there particular patient subgroups or tumour characteristics for whom surgery may provide a greater long-term oncologic advantage, and if so, is this sufficiently highlighted? he assertion that "the COLLISION trial reported a superior local tumour control rate for TA (by target tumour) compared to resection, establishing TA as the preferred treatment for small tumours amenable to ablation with margins" (Page 10) seems to be an excessively robust conclusion, particularly in light of the non-inferiority design. uld the authors clarify how a "better local tumour control rate" directly informs a preferable therapy decision within the framework of a non-inferiority trial, particularly considering the potential impact of factors like as patient selection or follow-up duration on this interpretation?
We sincerely thank the reviewer for this thoughtful and important comment. We would like to clarify that the intention of this review was not to suggest superiority of interventional approaches over surgical resection for colorectal liver metastases. On the contrary, we have incorporated and discussed numerous studies demonstrating the strong association of surgical resection with favorable local tumor control and long-term outcomes.
Within the manuscript, recurrence rates and overall survival data available in the literature are reported in detail, with most studies presenting outcomes over a two-year follow-up period. Given that the review already provides an extensive description of the technical aspects of each treatment modality, we felt that adding further detailed reporting of specific outcomes would go beyond the scope of this article. For that reason, we cited the corresponding articles that comprehensively provide these data.
The primary focus of this review is to highlight advances in interventional oncology and to illustrate how the interventional community has sought to integrate prognostic and technical factors long established in surgery, thereby aiming to achieve improved outcomes analogous to resection. This is highlighted on the first paragraph of the section 2.3:
“Evolution of Thermal Ablation as an Equivalent Treatment to Surgical Resection for Colorectal Liver Metastases ≤ 3 cm”.
More precisely: “Risk factors for local tumor progression (LTP) after resection include suboptimal surgical margins, residual viable tumor, ≥ 3 target tumors, CLM maximal longitudinal diameter ≥ 5 cm and carcinoembryonic antigen (CEA) levels > 200 ng/mL. The ability to perform pathologic assessment of the surgical margins constituted an objective advantage of surgery over percutaneous ablation. The realization that complete tumor resection with margins impacted oncologic outcomes, sparked an interest to assess the results of percutaneous ablation with a similar approach including confirmation of complete ablation with margins”
Following this rationale, we organized the manuscript into subunits focused on factors that interventional oncologists, in parallel with the surgical community, have sought to refine: Target tumor size, metastasis location, number and chemotherapy synergy, ablation margin assessment (= analogous of surgical margin assessment), optimization of ablation monitoring with split-dose PET/CT and CTHA, biopsy confirmation, patient eligibility for ablation, and establishment of a Clinical Risk Score. Each section highlights how interventional practices have mirrored surgical strategies in their evolution.
For example, at the section 2.3 and subunit 2.3.7: “Patient eligibility for ablation and establishment of a Clinical Risk Score”,
we highlighted that the modified clinical risk score made according to the surgical risk score: “At 2011 a pivotal single center study established a 4-scale modified clinical risk score (mCRS) for patients undergoing curative intent ablation for recurrent CLM following surgical resection. Similar to the surgical clinical risk score (CRS), mCRS contained specific disease characteristics associated with high predictive value for procedural outcomes. Lymph node-positivity, disease free survival less than 12 months, ≥ 1 CLM, longitudinal diameter of the larger tumor > 3cm (instead of 5 cm for surgical CRS) constituted the clinical factors and each one counts as 1 point towards the mCRS; CEA level was not included at that time-point due to lack of the corresponding information. High risk individuals (mCRS ≥ 3) were associated with 3.13 times greater likelihood of LTP in comparison to low risk patients (mCRS ≤ 2). The reported median overall survival was 21 months for the former group and 35 months for the latter group, respectively. In 2016 the same group reproduced the study for patients with unresectable CLMs or recurrent metastases after metastectomy, with the addition of CEA level > 30 ng/mL (instead of 200 in surgical CRS) as a prognostic factor for LTP and component of mCRS. According to the new 5-scale mCRS, high-risk patients were considered the ones with total score of 4 or 5. Tumor size > 3cm and ≥ 1 site of extrahepatic disease (EHD) retained statistical significance after multivariate analysis and presented an inverse relationship with overall survival, while the same relationship was reported for tumor size > 3cm and MM ≤ 5 mm with LTP free survival”.
In summary, our intention was to contextualize advances in interventional oncology within a framework that acknowledges the foundational role of surgery, while emphasizing how interventional practices have progressively incorporated similar prognostic principles to improve therapeutic outcomes.
Furthermore, regarding your question about the prognostic factors within the patient subgroup associated with the most favorable oncologic outcomes following thermal ablation, we have included a dedicated section 2.3.7 addressing this. Please see below:
“Patient eligibility for ablation and establishment of a Clinical Risk Score”
Establishment of the prognostic factors for oncologic outcomes was another important step in the evolution of ablation. Initial studies demonstrated that tumor size and lymph node positivity serve as independent prognostic factors for worse clinical outcome. At 2011 a pivotal single center study established a 4-scale modified clinical risk score (mCRS) for patients undergoing curative intent ablation for recurrent CLM following surgical resection. Similar to the surgical clinical risk score (CRS), mCRS contained specific disease characteristics associated with high predictive value for procedural outcomes. Lymph node-positivity, disease free survival less than 12 months, ≥ 1 CLM, longitudinal diameter of the larger tumor > 3cm (instead of 5 cm for surgical CRS) constituted the clinical factors and each one counts as 1 point towards the mCRS; CEA level was not included at that time-point due to lack of the corresponding information. High risk individuals (mCRS ≥ 3) were associated with 3.13 times greater likelihood of LTP in comparison to low risk patients (mCRS ≤ 2). The reported median overall survival was 21 months for the former group and 35 months for the latter group, respectively. In 2016 the same group reproduced the study for patients with unresectable CLMs or recurrent metastases after metastectomy, with the addition of CEA level > 30 ng/mL (instead of 200 in surgical CRS) as a prognostic factor for LTP and component of mCRS. According to the new 5-scale mCRS, high-risk patients were considered the ones with total score of 4 or 5. Tumor size > 3cm and ≥ 1 site of extrahepatic disease (EHD) retained statistical significance after multivariate analysis and presented an inverse relationship with overall survival, while the same relationship was reported for tumor size > 3cm and MM ≤ 5 mm with LTP free survival.”
Given that our review focuses on the evolution of interventional approaches for colorectal liver metastases, we believe that dedicating a section to the surgical resection subgroup of patients who might benefit falls outside the scope of this review, as it does not pertain to our subject.
One of the main limitation of the previous literature was the selection bias between ablation and resection of colorectal liver metastases cases was the selection bias derived from the fact that the cases that were amenable to thermal ablation was the ones rendered unresectable.
The COLLISION Trial represents an international, phase III randomized controlled trial that tried to address this selection bias. Our statement of better local tumor control rates is based on the authors statistically significant results. More precisely according to their published results at the publication “Thermal ablation versus surgical resection of small-size colorectal liver metastases (COLLISION): an international, randomised, controlled, phase 3 non-inferiority trial”, the authors report: “for per-tumour local control, the experimental group was superior (HR 0·09, 95% CI 0·01–0·74; p=0·024) to the control group.”. The reason of early cessation of this trial was due to reaching the predefined rules on predefined stopping rules: “a conditional likelihood to prove non-inferiority for the primary endpoint overall survival of more than 90% (91%), a superior safety profile, and a non-inferior local control for the experimental group”. The impact of COLLISION Trial in addressing this subject is depicted in the change of the international guidelines, as indicated below:
NCCN2024: “Resection is preferred over locally ablative procedures (eg, image-guided thermal ablation or SBRT). However, these local techniques can be considered for liver or lung oligometastases (COL-C and COL-E)”.
NCCN2025: “Resection is preferred over locally ablative procedures (eg, image-guided thermal ablation or SBRT). However, these local techniques can be considered for liver or lung oligometastases (COL-C and COL-E). For small lesions (≤3cm), thermal ablation is equivalent to resection”.
To avoid any misunderstanding, we changed our manuscript according to your recommendation as follows:
The subunit 2.3 title: “Evolution of Thermal Ablation as an Equivalent Treatment to Surgical Resection for Colorectal Liver Metastases ≤ 3 cm.”
We also rephrased the Guideline change section as follows: “There was an expected change in the choice of first line local curative intent treatment for small CLM after the COLLISION trial. This RCT stopped early for meeting predefined criteria for non-inferiority in terms of overall survival. The trial demonstrated non-inferiority regarding overall survival and local tumor control, shorter hospitalization time and fewer complications with TA compared to surgical resection for small-size colorectal liver metastasis up to 3 cm. The assumption that TA should only be used for not optimally resectable colorectal liver metastases changed in the latest update of the guidelines and TA is now considered equivalent to surgical resection for small (≤ 3 cm) CLM (figure 4)”.
The evaluation delineates the indications for each treatment method in general terms (e.g., "small CLM suitable for ablation with sufficient margins," "more extensive disease that is not suitable for resection or ablation"). The intricacies of patient selection within these categories, particularly with multidisciplinary team (MDT) talks, patient performance status, comorbidities, and previous systemic medications, warrant more elaboration.
Question: For each interventional modality addressed, in addition to tumour size and resectability, what are the essential patient-specific factors (e.g., ECOG performance status, liver function, systemic chemotherapy regimen, genetic mutations such as KRAS/BRAF beyond their reference for margin evaluation) that influence the decision-making process in a practical multidisciplinary team setting? Publication indicates that "current guidelines recommend TA as a standalone therapy (table 1)," although Table 1 is absent from the supplied PDF. Evaluation of the guideline revisions is unfeasible without Table 1. Kindly furnish Table 1 and specify its source (e.g., NCCN, ESMO, or other national/international guidelines) along with the level of recommendation for each situation.
As previously mentioned, current guidelines, e.g., the NCCN guidelines, accept thermal ablation as a standalone curative-intent therapy for colorectal liver metastases smaller than 3 cm, provided that all disease can be ablated with uniform minimal ablation margins of 5 millimeters around the tumor. More specifically:
NCCN2025: “Resection is preferred over locally ablative procedures (eg, image-guided thermal ablation or SBRT). However, these local techniques can be considered for liver or lung oligometastases (COL-C and COL-E). For small lesions (≤3cm), thermal ablation is equivalent to resection”.
A0 ablation covering 5mm margins is the equivalent of R0 surgical excision.
Furthermore, the objective of this review is to present the factors that have contributed to the evolution of interventional oncology. The approach to each individual colorectal liver metastasis case by a multidisciplinary team to determine the optimal treatment strategy is a separate topic, better suited for a review focused on that subject.
For the factors influencing decision-making regarding patient eligibility for ablation with increased probability of achieving optimal oncologic outcomes, we have included a dedicated section 2.3.7, with the title: “Patient eligibility for ablation and establishment of a Clinical Risk Score.”
Moreover, although the genetic profile of a patient with colorectal liver metastasis influences their prognosis, it does not constitute an exclusion criterion for the procedure.
We believe that the table briefly presenting the currently available interventional oncology treatment options for colorectal liver metastases may aid the reader’s understanding of the text. However, we fully acknowledge your concern regarding potential frustration, and in accordance with your recommendation, we will exclude it from this review.
The manuscript briefly discusses the synergy between percutaneous TA and systemic chemotherapy (e.g., CLOCC trial). It similarly references TARE in conjunction with HAIP treatment. A more systematic and analytical examination of the appropriate sequencing of interventional therapies alongside systemic treatments (e.g., neo-adjuvant versus adjuvant, concurrent versus sequential) and the data for these combinations would improve the review.
Question: What are the principal factors and problems in developing clinical trials to explore the appropriate sequencing and combination of systemic chemotherapy with interventional oncology treatments for colorectal liver metastases (CLM)? What are the deficiencies in evidence concerning multi-modality techniques, and which sorts of studies are most essential to rectify these deficiencies?
We sincerely appreciate and thank the reviewer for this thoughtful comment. Although this is a highly interesting area of investigation, we believe that focusing on systemic chemotherapy is a topic more pertinent to medical oncology and somewhat beyond the scope of this review. Our aim was to concentrate on the innovation and evolution of interventional techniques rather than to provide a broad discussion on the overall management of colorectal liver metastasis, as the ablation was added as an additional treatment to patients that were already under chemotherapeutic coverage.
In line with your concern, after presenting the synergy among trans-arterial radioembolization and hepatic arterial infusion pump, we have highlighted the need for further investigation at section 3.4.3:
“Another subject of interest for TARE in CLM management is the synergy with hepatic arterial infusion pump (HAIP). Literature supports clinical outcomes improvements from the addition of radioembolization to HAIP chemotherapy. Investigators assessed the value of the addition of a single dose of 90Y through the HAIP in patients with extensive liver metastases. After a time-interval of four weeks selective arterial 5-FU chemotherapy was re-initiated. The results supported improvement of survival times, especially in patient with absence of extra-hepatic disease. Another retrospective review evaluating factors that affect the efficacy outcome of radioembolization in heavily pretreated patients reported prolongation of the overall survival in study participants who received additional HAIP-mediated chemotherapy after TARE. In a prospective phase I clinical trial assessing TARE efficacy and safety outcomes in heavily pretreated patients who experienced local tumor progression, additional HAIP chemotherapy after radioembolization resulted in 50% reduction of CEA levels. Current evidence does not prohibit reinitiation of pump use upon progression after TARE provided that liver function enzymes (LFTs) and bilirubin levels remain acceptable. It is reasonable to investigate ways to optimize the combination of these modalities in subsequent work. There is a recently announced RCT that intends to evaluate the combination of TARE and HAIP for unresectable HCC (NCT06867432). Lastly, the bibliography almost exclusively contains studies evaluating the effect of 90Y radioembolization with very sparse data for other isotopes. Evaluation of the efficacy of other radio-active microspheres can greatly contribute to the optimization of clinical outcomes of TARE on metastatic colorectal cancer to the liver”.
The review examines the advancement of imaging modalities for margin evaluation (2D versus 3D, intraprocedural versus post-ablation) and the function of PET/CT and CTHA in monitoring. It also references biomarkers such as CEA and KRAS status. A more unified section on incorporating modern imaging and molecular indicators into treatment planning, response assessment, and recurrence surveillance, rather than isolated references, would be advantageous.
Question: How do contemporary imaging modalities (e.g., improved MRI sequences, quantitative PET metrics, AI-enhanced image processing) presently or potentially enhance patient selection for targeted interventional therapies? Aside from KRAS, are there further developing molecular biomarkers (e.g., circulating tumour DNA, gene expression profiles) that can forecast responses to specific interventional therapies or inform their use, and if so, how could they be incorporated into clinical practice for colorectal liver metastases (CLM)?
We appreciate the reviewer’s suggestion. The importance of intraprocedural ablation monitoring is indeed one of the key evolutionary aspects in optimizing the procedure. Since the current study focuses on advances in interventional approaches for colorectal hepatic metastases, we considered it more appropriate to provide the reader with an in-depth discussion of factors affecting procedural outcomes without including extensive information beyond this scope. Imaging considerations before, during, and after interventions for this disease have been comprehensively addressed in our previous publications, which are more suitable for this topic.
The abstract and discussion mention "salvage therapy." A comprehensive examination of the long-term results for patients undergoing interventional therapies in a salvage context is necessary, notably regarding overall survival, quality of life, and the management of later recurrences, despite the initial benefits of these treatments being outlined.
Question: What specific problems arise in defining and assessing long-term outcomes (e.g., following lines of therapy, cumulative toxicity, overall survival) for patients receiving interventional oncology procedures as salvage therapy for colorectal liver metastases (CLM) in severely pre-treated populations? What is the standard time and method of follow-up for patients undergoing salvage interventional treatments, and how does this contrast with those receiving curative purpose therapies?
We appreciate the reviewer’s comment. By definition, trans-arterial radioembolization and trans-arterial chemoembolization approaches are utilized as salvage treatments. Accordingly, the section of the review addressing these methods focuses on salvage therapy and local tumor control. Considering your thoughtful comment, we have added the survival rates following thermal ablation in salvage settings at a new subsection as follows:
“Treatment efficacy in salvage settings
Thermal ablation has become an established and effective treatment option for recurrence following resection or previous thermal ablation. This approach has demonstrated survival rates comparable to those of salvage surgical resection. Reported overall survival rates for salvage thermal ablation at 1, 3, and 5 years range from 90.1% to 98.9%, 46.2% to 62.6%, and 34.8% to 42.3%, respectively”.
We sincerely appreciate the exceptional insights and thoughtful comments provided by the reviewer. This valuable guidance has significantly broadened our perspective on the manuscript, enhancing both its depth and quality. Moreover, the constructive feedback has offered us important direction not only for improving this publication but also for approaching future research with greater rigor and clarity. We are grateful for the reviewer’s expertise and support throughout this process.

Reviewer 2 Report
Comments and Suggestions for Authors
The paper entitled "Interventional Oncology for Colorectal Liver Metastases: From local cure tosalvage therapy" is an interesting and timely narrative review addressing the
evolving role of image-guided locoregional therapies for colorectal liver metástases. The authors conduct a comprehensive analysis of both thermal and non-thermal modalities currently available for the management of colorectal liver metastases. Particular emphasis is given to ablation, radioembolization, chemoembolization, and histotripsy. The review integrates evidence from recent clinical trials, providing an updated synthesis of therapeutic outcomes.
This manuscript is of high clinical relevance, as the majority of patients with colorectal liver metastases are not candidates for hepatic resection, and locoregional therapies are assuming an increasingly important role.
The discussion on ablation margins, patient selection, integration with adjuvant therapy, and the evolution of guidelines adds valuable insight for clinical practice.
Nevertheless, some remarks mus be carried out and some alterations must be performed:
Introduction:
- The numbering in the Introduction appears inconsistent, as the item ‘1’ is missing”
Sections
- Some sections appear to be overly detailed, leading to a certain degree of redundancy (e.g., Thermal Technique Evolution and Dosimetry). A more concise synthesis of these parts would improve the overall readability and focus of the manuscript.
- The inclusion of a table summarizing the described techniques would enhance clarity and provide a useful overview for the reader: indications, tumor size limits, advantages, limitations, evidence level.
Conclusion:
- The conclusion could be strengthened by emphasizing the future role of each modality, identifying existing gaps in the evidence—particularly regarding histotripsy—and providing suggestions for future research.
Figures:
-The figures are of high quality and constitute an added value to the manuscript, effectively illustrating the clinical examples discussed in the text.
- In Figure 4, the legend should include the definitions of the abbreviations used.
- The source of the figures included in the text should be specified.
I would like to address a few questions to the authors for clarification.
1. Could the authors better explain which clinical criteria are most relevant when deciding between thermal ablation, TARE, or TACE in patients with unresectable CLM?
2. As histotripsy remains an experimental technique, it would be valuable if the authors could clarify whether they consider the current evidence sufficient to support its inclusion in clinical guidelines.
3. What future directions do the authors consider most relevant for this topic? Which studies would they recommend? Could biomarkers of response or the integration of immunotherapy play a role?
Author Response
Response to the Editor and Reviewers
Biomedicines
RESPONSE TO COMMENTS
Ref.: Manuscript ID: biomedicines-3812108
Interventional Oncology for Colorectal Liver Metastases: From Local Cure to Salvage Therapy
Dear Dr. Manish Tripathi and Dr. Puneet Vij,
We would like to thank you and the reviewers for your thoughtful consideration of our manuscript. We believe that your comments were really meaningful towards improving our manuscript. We have addressed each comment in a point-by-point response below:
Reviewer’s 2 comments/recommendations:
Introduction:
- The numbering in the Introduction appears inconsistent, as the item ‘1’ is missing”.
We want to thank the reviewer for pointing this out. We have now applied the number “1” to the introduction section:
“1. Introduction”
Sections
- Some sections appear to be overly detailed, leading to a certain degree of redundancy (e.g., Thermal Technique Evolution and Dosimetry). A more concise synthesis of these parts would improve the overall readability and focus of the manuscript.
We sincerely thank the reviewer for this thoughtful comment and valuable suggestion. The reason certain sections of our manuscript are discussed more extensively than others is that different interventional radiology (IR) modalities have undergone varying degrees of development over the past two decades in the management of colorectal liver metastases.
Thermal ablation, for example, has now become a recommended standalone, curative‑intent treatment. In the most recent version of the NCCN guidelines, it is considered equivalent to surgical resection for target lesions with a maximal diameter of 3 cm. Our objective was therefore not only to provide readers with the available evidence but also to present the perspective of the interventional radiology community and highlight how improvements in technique have elevated thermal ablation to a status comparable with surgical standards of care. We consider it crucial to discuss in detail the factors that contributed to positioning thermal ablation at the forefront steps of management algorithms for colorectal liver metastases.
With regard to 90Y radioembolization and radiation segmentectomy, there is growing cumulative evidence demonstrating a statistically significant association between absorbed dose and local tumor control. Beyond addressing safety considerations through the mandatory mapping procedure (e.g., avoidance of extrahepatic perfusion), advances in dosimetry have significantly improved the precision of microparticle delivery to the intended target regions. In this context, the degree of dose delivery to perfused liver tissue is the distinguishing factor that defines radiation segmentectomy—now regarded as a potentially curative‑intent treatment—from radioembolization.
The relative emphasis given to each modality in our manuscript reflects the volume and depth of existing evidence in the published literature as well as the extent of discussion in official guideline reports. We believe that this approach allows readers, including clinicians and future researchers, to both appreciate the noteworthy advances in interventional oncology for colorectal liver metastases and to identify knowledge gaps where further innovation and research are most needed.
Finally, we consider this review to be one of the most detailed and comprehensive analyses to date of the various interventional modalities, even the least investigated ones like histotripsy or the least utilized like chemoembolization, available for managing colorectal liver metastases. This is reflected in the carefully structured format of the manuscript and the precision with which each subtopic is addressed and referenced.
- The inclusion of a table summarizing the described techniques would enhance clarity and provide a useful overview for the reader: indications, tumor size limits, advantages, limitations, evidence level.
We sincerely thank the reviewer for this thoughtful comment and fully agree that a summary table can be a useful tool for readers, as it highlights the key aspects of the manuscript in a clear and accessible manner. However, given that similar studies have already included tables of the type the reviewer suggests, we chose to focus instead on providing high‑quality, precise, and comprehensive figures. We believe that these images not only enhance understanding of the content but also help avoid redundancy and overlap with previously published work.
Conclusion:
- The conclusion could be strengthened by emphasizing the future role of each modality, identifying existing gaps in the evidence—particularly regarding histotripsy—and providing suggestions for future research.
We thank the reviewer for this suggestion. We have now rephrased the conclusion (section 5) according to your recommendations: Please see below:
“There has been continuous evolution in the role of interventional oncology for the treatment of CLM, both in the locally-curative and salvage settings. These minimally-invasive treatments can be used alone or in combination with other systemic or locoregional therapies to provide disease control. Thermal ablation is now recognized as a standalone curative intent treatment option, considered equivalent to surgical resection for metastases with a maximal diameter of 3 cm. Radiation segmentectomy is also employed with potentially curative intent in carefully selected patients with limited number of CLM not amenable to resection or ablation, and advancements in dosimetry may allow 90Y trans-arterial radioembolization to be incorporated earlier in management algorithms for colorectal liver metastases. Furthermore, emphasis should be placed on evaluating emerging modalities such as histotripsy, with thorough investigation of their safety and efficacy profiles, in order to define the patient subgroups most suitable for these novel treatments. Ongoing clinical trials aim to evaluate the role of each treatment modality in the treatment paradigm.”
Figures:
-The figures are of high quality and constitute an added value to the manuscript, effectively illustrating the clinical examples discussed in the text.
- In Figure 4, the legend should include the definitions of the abbreviations used.
We want to thank the reviewer for the kind words and the appreciation of our effort. We have now addressed the issue of figure 4 as you indicated. Please see below:
Figure 4. Thermal ablation evolution.
- The source of the figures included in the text should be specified.
We thank the reviewer for pointing this out. All images included in this review are derived from cases performed by one of the authors (Dr. Sofocleous or Dr. Sotirchos) at our institution. For this reason, we considered additional clarification regarding the source of each figure to be unnecessary.
I would like to address a few questions to the authors for clarification.
1.Could the authors better explain which clinical criteria are most relevant when deciding between thermal ablation, TARE, or TACE in patients with unresectable CLM?
Deciding on the most appropriate treatment approach for colorectal liver metastases depends on a variety of patient‑specific factors. One of the most important is the size of the target tumor. Lesions smaller than 3 cm are generally preferred for ablation, whereas larger lesions are more commonly managed with transarterial 90Y radioembolization (TARE). Thermal ablation is favored in this context because it is a well‑established, definitive, curative‑intent treatment associated with excellent oncologic outcomes when adequate ablation margins are achieved—ideally greater than 5 mm, and optimally ≥10 mm.
In cases where anatomic location presents technical challenges—such as proximity to major structures including the portal veins, pericardium, or abdominal organs—when protective maneuvers (e.g., hydrodissection) are not feasible, non‑thermal approaches such as irreversible electroporation (when full patient paralysis can be safely administered, since this is essential for the procedure) or TARE may be more appropriate. Similarly, in patients with bile duct dilatation, thermal ablation is generally avoided to minimize the risk of biloma or other biliary complications.
Laboratory parameters, such as platelet count, are also critical in determining a patient’s eligibility for ablation or TARE, given their impact on procedural safety. In cases of multifocal disease, TARE is often favored over ablation due to its potential long‑term benefits, whereas ablation is primarily used in the setting of oligometastatic disease. At many institutions, the upper limit for ablation is considered to be five metastases; however, in our institution, we typically offer ablation to patients with up to three active colorectal liver metastases. Importantly, multimodal treatment strategies are frequently applied to patients with oligometastatic disease. For example, in a recent case, a patient presented with a 3.2‑cm metastasis in the posterior liver sector (segments VI and VII) and a second 1.2‑cm lesion in segment VIII. The smaller lesion was successfully treated with ablation, while the posterior sector metastasis was scheduled for radiation segmentectomy.
Transarterial chemoembolization (TACE) is less commonly used for colorectal liver metastases compared to other modalities. This is primarily due to the hypovascular nature of these tumors, which limits the effective distribution of embolic material, unlike hypervascular tumors such as hepatocellular carcinoma. For this reason, TARE is often preferred, as its therapeutic effect relies on the distribution of 90Y microspheres at the periphery of the target tumor, which deliver targeted intra‑arterial radiation capable of eradicating tumor cells within the treatment zone.
For patients with oligometastatic liver disease and target lesions smaller than 3 cm, thermal ablation is the preferred treatment option. When thermal ablation is not feasible or technically amenable, transarterial 90Y radioembolization (TARE) is considered. Transarterial chemoembolization (TACE) is rarely used in this setting and, when applied, it is generally supplementary to TARE or other treatment approaches.
- As histotripsy remains an experimental technique, it would be valuable if the authors could clarify whether they consider the current evidence sufficient to support its inclusion in clinical guidelines.
Histotripsy is a novel treatment modality that has only recently begun to be applied in the management of colorectal liver metastases. While early studies are ongoing, both its efficacy and safety profiles remain insufficiently defined for this tumor type. Consequently, although current evidence supports the feasibility of this technique, its exact role within the therapeutic algorithm has yet to be established. For this reason, we chose to summarize all available data and ongoing clinical trials investigating histotripsy, without making assumptions about the specific patient subgroups that may derive the most benefit. This approach is consistent with the current stance of the NCCN guidelines, which also acknowledge the need for further investigation before definitive recommendations can be made
- What future directions do the authors consider most relevant for this topic? Which studies would they recommend? Could biomarkers of response or the integration of immunotherapy play a role?
In our view, the most critical priority for interventional oncology at its current stage is the adoption of standardized principles across procedures, coupled with dedicated research aimed at further refining and advancing the available treatment modalities for colorectal liver metastases.
For thermal ablation, the procedure is considered complete when uniform ablation margins of at least 5 mm surround the tumor. Intra‑procedural assessment is essential, facilitated by recent technological advances such as 3‑dimensional software capable of measuring ablation zones and the improved resolution of computed tomography imaging. These tools are pivotal for confirming adequate margins and ensuring the absence of residual disease.
For radioembolization, further optimization of dosimetry and the establishment of clear patient‑selection criteria—including appropriate dose thresholds to achieve optimal oncologic outcomes—are critical next steps. Currently, 90Y radioembolization is most commonly utilized after disease progression following failure of two lines of chemotherapy. Addressing technical and clinical eligibility issues may allow for its use at earlier stages in the treatment algorithm.
In addition, biomarkers with prognostic value, including those previously reported by the authors of this review, hold promise in guiding patient selection. Immunotherapy has also demonstrated remarkable results, and its integration into a multidisciplinary framework—alongside chemotherapy and interventional modalities—may lead to improved oncologic outcomes. Such integration may ultimately support the development of new clinical scoring systems and the definition of patient subgroups most likely to benefit from novel therapeutic combinations.
Nevertheless, we consider a detailed discussion of these emerging aspects to be beyond the scope of the present article, as they are more closely aligned with the focus of the medical oncology community.
We are truly grateful for the reviewer’s exceptional feedback and thoughtful recommendations. The insights provided have allowed us to view our work from a broader and more nuanced perspective, substantially improving the manuscript’s quality. The reviewer’s guidance extends beyond this study, offering valuable lessons that will inform and enhance the rigor of our future publications. We sincerely thank the reviewer for their important contribution and support.

Round 2
Reviewer 1 Report
Comments and Suggestions for Authors
The author responded to my comments.